

**The Role of Meteorological Conditions and Pollution Control**
**Strategies in Reducing Air Pollution in Beijing during APEC 2014 and**
**Parade 2015**
Pengfei Liang[1], Tong Zhu[1*], Yanhua Fang[1], Yingruo Li[1, 2], Yiqun Han[1], Yusheng Wu[1],
Min Hu[1], and Junxia Wang[1]
[1]SKL-ESPC and BIC-ESAT, College of Environmental Sciences and Engineering,
Peking University, Beijing, 100871, China
[2]Environmental Meteorology Forecast Center of Beijing-Tianjin-Hebei, China
Meteorological Administration, Beijing, 100089, China
[*]Correspondence to: Tong Zhu (tzhu@pku.edu.cn)
**Abstract**
To control severe air pollution in China, comprehensive pollution control
strategies have been implemented throughout the country in recent years. To evaluate
the effectiveness of these strategies, the influence of meteorological conditions on
levels of air pollution needs to be determined. We therefore developed a generalized
linear regression model (GLM) to establish the relationship between the concentrations
of air pollutants and meteorological parameters. Using the intensive air pollution
control strategies implemented during the Asia-Pacific Economic Cooperation Forum
in 2014 (APEC 2014) and the Victory Parade for the Commemoration of the 70[th]





Anniversary of the Chinese Anti-Japanese War and the World Anti-Fascist War in 2015
(Parade 2015) as examples, we estimated the role of meteorological conditions and
pollution control strategies in reducing air pollution levels in Beijing. During the APEC
(1 October to 31 December 2014) and Parade (1 August to 31 December 2015)
sampling periods, atmospheric particulate matter of aerodynamic diameter $\leq 2.5$ μm
($PM_{2.5}$) samples were collected and gaseous pollutants ($SO_2$, $NO$, $NO_x$, and $O_3$) were
measured online at a site in Peking University (PKU). The concentrations of all
pollutants except ozone decreased dramatically (by more than 20%) during both events,
compared with the levels during non-control periods. To determine the influence of
meteorological conditions on the levels of air pollution, we first compared the air
pollutant concentrations during days with stable meteorological conditions (i.e. when
the daily average wind speed (WS) was less than 2.50 m s$^{-1}$ and planetary boundary
layer (PBL) height was lower than 290 m). We found that the average $PM_{2.5}$
concentration during APEC decreased by 45.7% compared with the period before
APEC and by 44.4% compared with the period after APEC. This difference was
attributed to emission reduction efforts during APEC. However, there were few days
with stable meteorological conditions during Parade. As such, we were unable to
estimate the level of emission reduction efforts during this period. Finally, GLMs based
only on meteorological parameters were built to predict air pollutant concentrations,
which could explain more than 70% of the variation in air pollutant concentration levels,
after incorporating the nonlinear relationships between certain meteorological
parameters and the concentrations of air pollutants. Evaluation of the GLM





performance revealed that the GLM, even based only on meteorological parameters,
could be satisfactory to estimate the contribution of meteorological conditions in
reducing air pollution, and hence the contribution of control strategies in reducing air
pollution. Using the GLM, we found that the meteorological conditions and pollution
control strategies contributed 30% and 28% to the reduction of the $PM_{2.5}$ concentration
during APEC 2014, and 38% and 25% during Parade 2015. We also estimated the
contribution of meteorological conditions and control strategies implemented during
the two events in reducing the concentrations of gaseous pollutants and $PM_{2.5}$
components with the GLMs, revealing the effective control of anthropogenic emissions.

## 52 **1 Introduction**

Air pollution poses serious health risks to human populations and is one of the
most important global environmental problems. To control air pollution in China, the
State Council of China (2013) has released the Action Plan for Air Pollution Prevention
and Control, which sets pollution control targets for different regions, e.g. atmospheric
particulate matter of aerodynamic diameter ≤ 2.5 μm ($PM_{2.5}$) concentrations in 2017
shall fall in Beijing–Tianjin–Hebei (BTH) by 25%, in the Yangtze River Delta by 20%,
and in the Pearl River Delta by 15%, compared with 2012 levels. To meet these targets,
comprehensive pollution control strategies have been implemented at the national,
provincial, and city levels. However, it is not clear how effective these strategies are in
reducing air pollution. One of the challenges in evaluating the effectiveness of these
strategies is that the long-term strategies cannot improve air quality in the short term.





The efforts made to ensure satisfactory air quality for special events, such as the Beijing
2008 Olympics, provide a unique opportunity to evaluate the effectiveness of pollution
control strategies (Kelly and Zhu, 2016). During the Beijing Olympics comprehensive
pollution control strategies were implemented intensively over a short period of time.
Based on the successful experience during this event, the Chinese government
implemented similar air pollution control measures for the 41[st] Shanghai World Expo
in 2010 (SEPB, 2010), the 16[th] Guangzhou Asian Games and Asian Para Games in 2010
(GEPB, 2009), and the Chengdu Fortune Forum 2013 (CEPB, 2013). To ensure
satisfactory air quality in Beijing during the two most recent events: the Asia-Pacific
Economic Cooperation Forum in 2014 (APEC 2014) and the Victory Parade for the
Commemoration of the 70[th] Anniversary of the Chinese Anti-Japanese War and the
World Anti-Fascist War in 2015 (Parade 2015), the Chinese central government and the
local government in Beijing, together with its surrounding provinces, implemented
comprehensive air pollution control strategies, including the control of emissions from
traffic, industry, and coal combustion, as well as dust pollution control (Table 1). These
two events provide a good opportunity to evaluate the effectiveness of air pollution
control strategies.
One challenge when evaluating the effectiveness of air pollution control strategies
over a short period of time is separating out the contribution of meteorological
conditions from the reduction in air pollution levels.
Most previous studies have only provided a descriptive analysis of the changing
concentrations of air pollutants during these events. Wen et al. (2016) reported that the





average PM$_{2.5}$ concentration during APEC decreased by 54%, 26%, and 39% compared
with the levels before APEC in Beijing, Shijiazhuang, and Tangshan, respectively. The
authors also reported that the average concentration of total elements in PM$_{2.5}$ during
APEC decreased by 75%, 35%, and 36% compared with the levels before APEC in
these three sites, respectively. Han et al. (2015) observed that the extinction coefficient
and absorbance coefficient decreased significantly during APEC compared with the
values before.

An increasing number of studies have recognized the importance of

meteorological conditions in determining air pollution in Beijing and North China Plain
(e.g., Zhang et al., 2012). A northerly wind is considered to be favourable for pollutant
diffusion, while a southerly wind is considered to be favourable for the transport of
pollutants to Beijing (Zhang et al., 2014). When assessing the effectiveness of air
pollution control strategies, a few studies have distinguished between the contribution
of meteorological conditions and pollution control strategies in reducing air pollution
by comparing air pollutant concentrations under similar meteorological conditions
(Wang et al., 2015; Zhang et al., 2009). However, in these studies, days with stable
meteorological conditions were determined subjectively, which may introduce
uncertainties and inconsistencies when estimating changes in air pollutant
concentrations.

Statistical models have been developed to establish the relationship between air

pollutant concentrations and meteorological parameters. Table 2 summarizes these
models, with their respective R$^2$ values. Multiple linear regression models have been





widely applied to demonstrate the quantitative relationship between air pollutant
concentrations and meteorological parameters, by assuming a linear relationship.
However, these relationships are often non-linear (Liu et al., 2007; Liu et al., 2012).

Table 2 shows that most of the models with good explanation ($R^2 > 0.6$) have

actually adopted visibility, aerosol optical depth (AOD), and air quality index (AQI) as
independent variables to improve the performance of the regression models (Liu et al.,
2007; Sotoudeheian and Arhami, 2014; Tian and Chen, 2010; You et al., 2015). This
could cause problems in the prediction of air pollutant concentrations during intensive
emission control periods because visibility, AOD, and AQI are also dependent on air
pollution levels; hence, the statistical models may not function when air pollutant levels
are drastically reduced over a short period. A statistical model based solely on
meteorological parameters to predict air pollutant concentrations is therefore required.

In this study, we used the air pollution control periods during APEC 2014 and

Parade 2015 to estimate the role of meteorological conditions and pollution control
strategies in reducing air pollution in the megacity of Beijing. We first measured the
changes in air pollutant concentrations, including $PM_{2.5}$, gaseous pollutants, and the
components of $PM_{2.5}$. We then estimated the role of meteorological conditions and
pollution control strategies in reducing air pollution by comparing the pollutant
concentrations during days with stable meteorological conditions. Finally, we
developed a statistical model based only on meteorological parameters to evaluate the
role of meteorological conditions and pollution control strategies in reducing the levels
of air pollution in Beijing. Compared with the models used in previous studies, our





statistical model had the following advantages: (1) all of the independent variables were
meteorological parameters; (2) we considered the non-linear relationships between air
pollutant concentrations and meteorological parameters; and (3) in addition to
predicting PM$_{2.5}$ mass concentrations, our model could also predict concentrations of
gaseous pollutants and individual PM$_{2.5}$ components.
**2 Measurements and Methods**
**2.1 Measurements of Air Pollutants**
Gaseous pollutants (SO$_2$, NO, NO$_x$, and O$_3$) were measured online, and PM$_{2.5}$
samples were collected on filters at an urban monitoring station in the campus of Peking
University (39.99°N, 116.33°E) northwest of Beijing (Huang et al., 2010). The station
is located on the roof of a six-floor building, about 20 m above the ground and about
550 m north of the fourth ring road.
A PM$_{2.5}$ four-channel sampler (TH-16A, Wuhan Tianhong Instruments Co., Ltd.,
Hubei, China) was used to collect PM$_{2.5}$ samples. The sampling duration was 23.5 h
(from 09:30 to 09:00 LT the next day). Both 47-mm quartz filters (QM/A, Whatman,
Maidstone, England) and Teflon filters (PTFE, Whatman) were used. The flow rate was
calibrated to 16.7 L min$^{-1}$ each week and a blank PM$_{2.5}$ sample was collected once a
month. The quartz filters were baked at 550°C for 5.5 h before use. Immediately after
collection, the filter samples were stored at −25°C until analysis. A total of 225 PM$_{2.5}$
filter samples were collected during APEC (1 October to 31 December 2014) and
Parade (1 August to 31 December 2015) sampling periods. During the sampling periods,



20 days of PM$_{2.5}$ samples were missed due to rain or sampler failures. Sulphur dioxide
(SO$_2$) was measured with an SO$_2$ analyzer (43i TL, Thermo Fisher Scientific, Waltham,
MA, USA), with a precision of 0.05 ppb. Nitric oxide (NO) and nitrogen oxides (NO$_x$)
were measured with a NO-NO$_x$ analyzer (42i TL, Thermo Fisher Scientific), with
precisions of 0.05 ppb for NO and 0.17 ppb for NO$_2$. Ozone (O$_3$) was measured with
an O$_3$ analyzer (49i, Thermo Fisher Scientific), with a precision of 1.0 ppb. The SO$_2$
and NO-NO$_x$ analyzers both had a detection limit of 0.05 ppb, and the O$_3$ analyzer had
a detection limit of 0.50 ppb. All of the gaseous pollutant analyzers had a time
resolution of 1 min, and were maintained and calibrated weekly following the
manufacturer's protocols.
**2.2 Meteorological Data**
Meteorological data were obtained from the National Climate Data Center
(www.ncdc.noaa.gov) dataset. The meteorological parameters were monitored at a
station located in the Beijing Capital International Airport, and consisted of temperature
(T), relative humidity (RH), wind direction (WD), wind speed (WS), sea level pressure
(SLP), and precipitation (PREC). The PBL height was computed from the simulation
results of the National Center for Environmental Prediction (NCEP) Global Data
Assimilation System (GDAS) model (www.ready.arl.noaa.gov/READYamet.php).
**2.3 Analysis of the PM$_{2.5}$ Filter Samples**
To obtain daily average PM$_{2.5}$ mass concentrations, Teflon filters were weighed
before and after sampling using an electronic balance, with a detection limit of 10 µg



(AX105DR) in a super-clean lab (T: $20 \pm 1°C$, RH: $40 \pm 3\%$). A portion of each Teflon
filter was extracted with ultrapure water for the measurement of water-soluble ions ($Na^+$,
$NH_4^+$, $K^+$, $Mg^{2+}$, $Ca^{2+}$, $SO_4^{2-}$, $NO_3^-$, and $Cl^-$), with an ion-chromatograph (IC-2000 &
2500, Dionex, Sunnyvale, CA, USA). The detection limits of $Na^+$, $NH_4^+$, $K^+$, $Mg^{2+}$,
$Ca^{2+}$, $SO_4^{2-}$, $NO_3^-$, and $Cl^-$ were 0.03, 0.06, 0.10, 0.10, 0.05, 0.01, 0.01, and 0.03 mg
$L^{-1}$, respectively. A portion of each Teflon filter was digested with a solution consisting
of nitric acid ($HNO_3$), hydrochloric acid ($HCl$), and hydrofluoric acid ($HF$) for the
measurement of trace elements (Na, Mg, Al, Ca, Mn, Fe, Co, Cu, Zn, Se, Mo, Cd, Ba,
Tl, Pb, Th and U), with inductively coupled plasma-mass spectrometry (ICP-MS,
Thermo X series, Thermo Fisher Scientific). The recoveries for all measured elements
fell within $\pm 20\%$ of the certified values. A semi-continuous organic carbon/elemental
carbon (OCEC) analyzer (Model 4, Sunset Laboratory, Tigard, OR, USA) was used to
analyze organic and elemental carbon from a round punch (diameter: 17 mm) from each
quartz filter sample. The T protocol of the National Institute for Occupational Safety
and Health (NIOSH) thermal-optical method was applied (see details in Table S1).
All analytical instruments were calibrated before each series of measurements. The
$R^2$ values of the calibration curves for ions, elements, and sucrose concentrations were
higher than 0.999.

**2.4 Generalized Linear Regression Model (GLM)**

A generalized linear regression model (GLM) was used to establish the
relationship between air pollutant concentrations and meteorological parameters. The





objective dependent variables included concentrations of $PM_{2.5}$, individual $PM_{2.5}$
components, and gaseous pollutants.

To match the 23.5-h (09:30–09:00 LT the next day) sampling time of the $PM_{2.5}$

filter samples, metrological parameters were averaged over the same time span (Table
3) and used in the GLM alongside other parameters, e.g. the daily maximum of certain
meteorological parameters. The meteorological parameters used in the GLM were T,
RH, WD, WS, PBL height, SLP, and PREC. WDs were grouped into three categories,
with relevant values and assigned to each category: north (NW, W and NE) as 1, south
(SW, SE and E) as 2, and "calm and variable" as 3. A calm wind was defined as when
the WS was less than 0.5 m s$^{-1}$. A variable WD was defined as a condition when: (1)
the WD fluctuated by 60° or more during a 2-min evaluation period, with a WS greater
than 6 knots (11 km h$^{-1}$); or (2) the WD was variable and the WS was less than 6 knots
(11 km h$^{-1}$).

A preliminary analysis showed that the concentrations of air pollutants and

meteorological parameters fitted best with an exponential function or power function
(Figure S2); therefore, these functions were natural log transformed and introduced into
the GLM.

We applied the stepwise method to evaluate the level of multicollinearity between

the independent variables based on relevant judgement indexes, such as the variance
inflation factor (VIF) or tolerance. Based on the assumption that the regression residuals
followed a normal distribution and homoscedasticity, which is discussed in a later
section, we developed the following model to calculate the concentrations of air





pollutants and chemical components of $PM_{2.5}$ based on meteorological parameters:
$$ln\,C_{ij} = \beta_0 + \sum_{k=1}^{m} \beta_{1k} x_k + \sum_{k=1}^{n} \beta_{2k}\,ln\,x_k + \sum_{k=1}^{m'} \beta_{3k} x_k\,(lag) + \sum_{k=1}^{n'} \beta_{4k}\,ln\,x_k\,(lag) \qquad (1)$$
where $C_{ij}$ is the concentration of the $j^{th}$ air pollutant averaged over the $i^{th}$ day, $x_k$ is the
$k^{th}$ meteorological parameter, $\beta_k$ is the regression coefficient of the $k^{th}$ meteorological
parameter, and $\beta_0$ is the intercept. For meteorological parameters containing both
positive and negative values (i.e. T), only the exponential form was applied. $m$, $n$, $m'$,
and $n'$ are the number of different forms of meteorological parameters that were
eventually included in the model, and were determined based on the stepwise entering
method of the regression model. The suffix of *(lag)* refers to the meteorological
parameters of the previous day. The main assumption for equation (1) was that the
concentrations of air pollutants were only a function of the meteorological parameters,
and the emission intensities were constant. Hence, we only used the data before and
after APEC 2014 and Parade 2015 control periods in equation (1), excluding the data
collected during each period and during the heating season, e.g. after 15 November

2014.

**3 Results and Discussion**
**3.1 Changes of Air Pollutant Concentrations during the APEC 2014 and Parade**
**2015 Campaigns**
Figure 1 shows the time series of $PM_{2.5}$ and the concentrations of its components,
as well as the meteorological parameters during the APEC 2014 and Parade 2015
campaigns. The APEC 2014 campaign consisted of three distinct periods: before APEC





(BAPEC, 18 October to 2 November 2014), during APEC (APEC, 3 to 12 November
2014), and after APEC (AAPEC, 13 to 22 November 2014). The Parade 2015 campaign
was also divided into three distinct periods: before Parade (BParade, 1 to 19 August
2015), during Parade (Parade, 20 August to 3 September 2015), and after Parade
(AParade, 4 to 23 September 2015).

There were two pollution episodes during APEC, on 4 November and 7–10

November 2014, which corresponded to two relatively stable periods with low WS,
mainly from the south. The T declined gradually from 12.2°C before APEC to 4.9°C
after APEC, and the RH was above 60% during the two pollution episodes. During
Parade, the $PM_{2.5}$ concentrations were low during the whole control period, with the
prevailing WD from the north and low WS. The T was mostly higher than 20°C, which
differed from that during the APEC campaign when it was lower than 20°C.

Table 4 lists the mean concentrations and standard deviations of $PM_{2.5}$, gaseous

pollutants, and $PM_{2.5}$ components during the APEC and Parade campaigns. The mean
concentration of $PM_{2.5}$ during APEC was $48 \pm 35$ µg m$^{-3}$, 58% lower than during
BAPEC ($113 \pm 62$ µg m$^{-3}$), and 51% lower than during AAPEC ($97 \pm 84$ µg m$^{-3}$). The
mean concentration of $PM_{2.5}$ during Parade was $15 \pm 6$ µg m$^{-3}$, 63% lower than during
BParade ($41 \pm 14$ µg m$^{-3}$), and 62% lower than during AParade ($39 \pm 28$ µg m$^{-3}$).

Figure 1 here

Figure 2 shows the proportion of the measured $PM_{2.5}$ components, including OC;

EC; the sum of the sulphate, nitrate, and ammonia (SNA); and chloride ion (Cl$^-$) and
trace elements, which together accounted for 70–80% of the total $PM_{2.5}$ mass





concentration. The proportions of OC (23.5%) and EC (3.5%) in $PM_{2.5}$ were highest
during APEC. The proportion of SNA in $PM_{2.5}$ during APEC (40.6%) was lower than
during BAPEC (50.7%) and higher than during AAPEC (37.2%). The proportions of
$Cl^-$ (4.3%) and elements (6.8%) in $PM_{2.5}$ during APEC were higher than during BAPEC
and lower than during AAPEC. For the Parade campaign, the proportions of OC (26.6%)
and elements (6.6%) in $PM_{2.5}$ were highest during Parade. The proportions of EC (4.9%)
and $Cl^-$ (1.1%) in $PM_{2.5}$ during Parade were higher than during BParade and lower than
during AParade. The proportion of SNA in $PM_{2.5}$ was lowest during Parade (37.3%).
Similarly, during the pollution control periods of APEC and Parade, the proportions of
OC and elements in $PM_{2.5}$ tended to increase and the proportion of SNA in $PM_{2.5}$ tended
to decrease.

Figure 2 here

EC is usually considered to be a marker of anthropogenic primary sources, while

the sources of OC include both primary and secondary organic aerosols. The correlation
between OC and EC can reflect the origin of carbonaceous fractions (Chow et al., 1996).
Figure 3 shows the correlation between EC and OC concentrations during the APEC
and Parade campaigns. During the APEC and Parade campaigns, the correlation
coefficient during both control periods ($R^2 = 0.9032$) was larger than that during non-
control periods ($R^2 = 0.6468$), indicating that OC and EC were mainly derived from the
same sources during both pollution control periods, and were from different sources
during the non-control periods. The slope of the OC/EC correlation during the pollution
control period was 6.86, which was higher than that during the non-control period (3.97).



This could be due to high levels of secondary OC (SOC) formation during the control
periods, and/or the higher contribution from residential solid fuel (coal and biomass)
burning (Liu et al., 2016).

Figure 3 here

Figure 4 shows the proportion of SNA in $PM_{2.5}$ ($\rho(SNA)/PM_{2.5}$), the sulphur (S)
oxidation ratio (SOR = $[SO_4^{2-}]/([SO_2]+[SO_4^{2-}])$), and nitrogen oxidation ratio (NOR =
$[NO_3^-]/([NO_x]+[NO_3^-])$), along with $PM_{2.5}$ concentrations during the APEC (a) and
Parade (b) campaigns. During APEC, the average $\rho(SNA)/PM_{2.5}$ was 27%, which was
significantly lower than during BAPEC (42%). During Parade, the average
$\rho(SNA)/PM_{2.5}$ was 35%, which was also significantly lower than during BParade (47%).
During the APEC campaign, the average $SO_2$ concentration was 11.3 µg m$^{-3}$
before APEC, 9.5 µg m$^{-3}$ during APEC, and 34.8 µg m$^{-3}$ after APEC, respectively. The
average $NO_x$ concentration was 151 µg m$^{-3}$ before APEC, 81 µg m$^{-3}$ during APEC, and
220 µg m$^{-3}$ after APEC, respectively. During the Parade campaign, the average $SO_2$
concentration during Parade was 1.6 µg m$^{-3}$, lower than both during BParade (2.7 µg
m$^{-3}$) and AParade (5.9 µg m$^{-3}$). The average $NO_x$ concentration was also lower during
Parade (26 µg m$^{-3}$), than during BParade (57 µg m$^{-3}$) and AParade (63 µg m$^{-3}$).
During the APEC campaign, both the SOR and NOR declined gradually. The
average SOR was 42%, 27%, and 17% in the BAPEC, APEC, and AAPEC periods,
respectively. The average NOR was 13%, 8%, and 5% in the BAPEC, APEC, and
AAPEC periods, respectively. SOR and NOR exhibited different patterns during the
Parade campaign. The average SOR was 75%, 64%, and 55% in the BParade, Parade,




and AParade periods, respectively. The average NOR was 8%, 5%, and 8% in the
BParade, Parade, and AParade periods, respectively. The SOR was higher during the
Parade campaign (64%) than during the APEC campaign (30%). For NOR, a higher
average value was found during the APEC campaign (9%) than during the Parade
campaign (7%).
The APEC campaign occurred during autumn and early winter, while the Parade
campaign occurred during late summer and autumn. The active photochemical
oxidation during the Parade campaign resulted in high $SO_2$-to-sulphate transformation
rates, as indicated by the high SOR. In addition, the higher RH in summer favoured the
heterogeneous reaction of sulphate formation. For NOR, the T was higher during Parade
than during APEC, which favoured the volatilization of nitric acid and ammonia from
the particulate phase of nitrate.
These results indicate significant reductions of air pollution during the pollution
control periods of APEC 2014 and Parade 2015. However, it is necessary to evaluate if
meteorological conditions contributed to this improvement.
Figure 4 here
**3.2 Reduction of Air Pollution under Similar Meteorological Conditions**
Figure 5 shows the prevalence of WD during the APEC and Parade campaigns.
During APEC, the prevailing WD was from the north and northwest, and accounted for
30–40% of the wind frequency. The mean WS during APEC was 3.1 m s$^{-1}$, higher than
during BAPEC (2.2 m s$^{-1}$) and AAPEC (2.4 m s$^{-1}$). The "calm and variable" proportion




of APEC was 28.5%, which was lowest during the APEC campaign. During Parade, a
northern and northeastern WD accounted for more than 30% of the wind frequency, and
the "calm and variable" proportion was 18.5%, much lower than during BParade and
AParade (25.7% and 20.3%, respectively).

Figure 5 here

Figure 6 shows a time series of daily average $PM_{2.5}$ concentrations and PBL

heights during the APEC and Parade campaigns, indicating that they have an anti-
correlation. In both the BAPEC and AAPEC periods, the PBL heights were mostly less
than 400 m. Compared with during APEC and during Parade, the PBL heights were
constantly high, which was much more favourable for the diffusion of air pollutants
during the control period.

Both WS and PBL height during APEC and Parade were favourable for pollutant

diffusion. Therefore, it is necessary to consider meteorological conditions when
assessing the impacts of pollution control. One way to do this is to compare air pollution
concentrations during periods when meteorological conditions were the same, i.e. under
stable conditions (Wang et al., 2015; Zhang et al., 2009).

Figure 6 here

**3.2.1 Identify Stable Meteorological Periods**

Stable conditions can be defined based on the relationship between air pollution

levels and both WSs and PBL height. Figure 7 shows scatter plots between $PM_{2.5}$
concentrations and WS and PBL heights. The relationship can be fitted with a power



function. A stable condition could be defined by identifying the turning points when the
slopes changed from large to relatively small values, and stable conditions could be
defined when WSs and PBL heights were lower than the values of the turning points.

The slopes of the power function were monotone, varying with no inflection point.

Thus, we used piecewise functions to identify the turning points. As Figure 7 shows,
the intersections of two fitting lines represented the turning points of the meteorological
influence on $PM_{2.5}$; thus, we defined days with stable meteorological conditions to be
those with a daily average WS less than 2.50 m s$^{-1}$ and a daily average PBL height
lower than 290 m. We could then compare the corresponding pollutant concentrations
between days with stable meteorological conditions.

Figure 7 here

**3.2.2 Variation of Air Pollutant Concentrations under Stable Meteorological**
**Conditions**

The days with stable meteorological conditions were determined with the method

introduced in Section 3.2.1. As a result, eight days before APEC (18, 22, 23, 24, 28, 29,
30, and 31 October 2014), six days during APEC (3, 4, 7, 8, 9, and 10 November 2014),
and seven days after APEC (14, 15, 17, 18, 19, 20, and 22 November 2014) were
defined as having stable meteorological conditions. Table 5 lists the meteorological
conditions (WSs and PBL heights), and the concentrations of pollutants on the days
with stable meteorological conditions during the APEC campaign. For the Parade
campaign, only one day in each of the BParade, Parade, and AParade periods was
defined as having stable meteorological conditions. This was considered to not be well





representative of the Parade campaign. Thus, we only assessed the variation of air
pollutant concentrations during stable meteorological periods of the APEC campaign.

For days with stable meteorological conditions during the APEC campaign, the

average WS was 1.4, 1.9, and 1.7 m s$^{-1}$ in the BAPEC, APEC, and AAPEC periods,
respectively; and the average PBL height was 191, 194, and 160 m in the same three
periods, respectively. This clearly shows that the meteorological conditions of days
considered to be stable throughout the APEC campaign were very similar.

Figure 8 shows the percentage reductions calculated by comparing the decreased

average concentrations for all days during APEC to the average concentrations before
APEC in black bars, and the percentage reductions based on the days with stable
meteorological conditions in red bars. For the difference between the APEC and
BAPEC periods, the percentage reduction on days with stable meteorological
conditions was much lower than the reduction calculated when considering all days,
except for Ca and NO. This indicates that the method applied to days with stable
meteorological conditions excluded part of the meteorological influence on pollutant
concentrations.

The standard deviations were also calculated with an error transfer formula that is

described in detail in the Supplementary Information. Figure 8 shows that the standard
deviations of the percentage reduction based on days with stable meteorological
conditions decreased significantly. For example, the standard deviation of the
percentage reduction in PM$_{2.5}$ based on the days with stable meteorological conditions
decreased from 39% to 26% compared with the same measurement when all days were



considered. This indicates that by considering only days with stable meteorological
conditions, the uncertainties associated with the percentage reduction figures were
reduced and the reliability of the changes of air pollutants concentrations were
improved.
Figure 8 here
Figure 9 shows the changes of pollutant concentrations on days with stable
meteorological conditions during the APEC campaign. The average $PM_{2.5}$
concentration was 70 μg m$^{-3}$ during APEC, which represented a 45.7% decrease
compared with the concentration in the BAPEC period (129 μg m$^{-3}$) and a 44.4%
decrease compared with the concentration in the AAPEC period (126 μg m$^{-3}$).
A similar pattern was observed for SNA. The SNA concentration decreased
significantly during APEC compared with the BAPEC period. In the BAPEC period,
the average sulphate, nitrate, and ammonium concentrations were 13.4, 34.2, and 16.7
μg m$^{-3}$, respectively. During APEC, the average concentrations of sulphate, nitrate and
ammonium were 5.6, 17.0, and 7.4 μg m$^{-3}$, respectively. In the AAPEC period, the
average concentrations of sulphate, nitrate, and ammonium were 12.5, 21.8, and 13.6
μg m$^{-3}$, respectively.
The average OC concentration was 15.9 μg m$^{-3}$ during APEC, which represented
a 15.4% decrease compared with the concentration in the BAPEC period (18.8 μg m$^{-3}$).
In comparison, the average EC concentration was 2.4 μg m$^{-3}$ during APEC, which
represented a 35.1% decrease compared with the concentration in the BAPEC period
(3.7 μg m$^{-3}$). This indicates that the reduction of the OC concentration was less



significant than that of the EC concentration during APEC.
During APEC, the average concentrations of $Cl^-$ and potassium ion ($K^+$) were 3.1
and 0.96 μg m$^{-3}$, which represented a decrease of 22.5% and 31.4%, compared with the
concentrations in the BAPEC period. The average concentrations of $Cl^-$ and $K^+$
increased significantly to 8.9 and 2.01 μg m$^{-3}$ in the AAPEC period, which represented
an increase of 187.1% and 109.4%, respectively compared with the concentrations
during APEC. The average concentrations of several anthropogenic elements, including
Pb, Zn, Ni, and Mn, all significantly decreased during APEC compared with the
concentrations in the BAPEC and AAPEC periods. In contrast, the average
concentration of Ca was 634 ng m$^{-3}$ during APEC, which represented only an 8.2%
decrease compared with the concentration before APEC (691 ng m$^{-3}$), this may be due
to the fact that Ca is mainly derived from geogenic sources (Mustaffa et al., 2014; Tao
et al., 2014).
The average $SO_2$ concentration was 12.7 μg m$^{-3}$ during APEC, which was almost
equal with that in the BAPEC period (12.9 μg m$^{-3}$), but it increased significantly to 41.8
μg m$^{-3}$ in the AAPEC period. In comparison, the average concentrations of NO and
$NO_x$ decreased significantly during APEC (61.0% and 42.9%, respectively) and
increased substantially in the AAPEC period (376.7% and 139.3%, respectively).
During APEC, the average concentration of $O_3$ was 27.1 μg m$^{-3}$, which represented an
increase of 92.2% and 133.6%, compared with the concentrations in the BAPEC and
AAPEC periods. The significant increase in the average $SO_2$ concentration in the
AAPEC period was consistent with the increased ratios of the average concentrations



432 of $Cl^-$ and $K^+$ in the same period, indicating an increase in coal combustion, which

433 coincided with the government subsidised heating season in north China that started on

434 15 November. The inverse variation patterns of $NO_x$ and $O_3$ indicate that the significant

435 increase of $O_3$ may be because of the decline in average $NO_x$ concentration during the

436 pollution control period of APEC.

437             Figure 9 here

438   Table 6 lists the percentage differences among the mean $PM_{2.5}$ concentrations of

439 four periods (P1, P2, P3, and P4) that were randomly selected from within the non-

440 control days of the APEC and Parade campaigns. Based on the assumptions that days

441 with stable meteorological conditions were representative of the corresponding periods

442 during the APEC campaign, and the emission intensities were constant, the percentage

443 differences in the mean $PM_{2.5}$ concentrations between these four random periods should

444 be close to zero. The mean concentrations during P1, P2, P3, and P4 were 120, 101, 96,

445 and 87 µg m$^{-3}$, respectively. The standard deviation (SD) during P1, P2, P3, and P4

446 were 97, 58, 40, and 23 µg m$^{-3}$, respectively, with the average SD being 59 µg m$^{-3}$. The

447 mean value of the percentage differences of the mean $PM_{2.5}$ concentrations between P1,

448 P2, P3, and P4 was −16%, with a root mean square error (RMSE) of 18%. Hence,

449 uncertainties remain within the percentage differences based on the days with stable

450 meteorological conditions, although the size of these uncertainties was reduced. This

451 may be due to the limited sample size on days with stable meteorological conditions

452 during the APEC campaign. It is therefore necessary to further quantify the

453 meteorological influences.



### 3.3 Emission Reductions during APEC and Parade Based on GLM Predictions


The previous section showed that the number of days with stable meteorological
conditions could be limited; it was therefore impossible to estimate quantitatively the
contribution of meteorological conditions to the reduction of air pollutant
concentrations. We developed a GLM based only on meteorological parameters to meet
this requirement.
**3.3.1 Model Performance**
Figure 10 shows the scatter plot and correlation between the GLM-predicted
and observed concentrations of air pollutants transformed to a natural log. The $R^2$ values
of the linear regression equations ranged from 0.6638 to 0.8542, most of them are
higher than 0.7 except for Zn and Mn, indicating that the GLM-predicted concentrations
correlated well with the observed concentrations. Specifically, the $R^2$ value of the linear
regression equation for $PM_{2.5}$ is as high as 0.8154.

Figure 10 here

Before applying the GLM to predict the air pollutant concentrations, the cross-
validation (CV) method was used to evaluate the performance of the $PM_{2.5}$ model, with
the assumption that it was representative of all air pollutants. The data input to the $PM_{2.5}$
model was allocated randomly into five equal periods, namely CV1, CV2, CV3, CV4,
and CV5. For each test, one period was removed from the input data and the remaining
data were applied to establish the CV model, which was then used to predict the $PM_{2.5}$
concentrations for the removed period. After five rounds, all input data were included



in the CV test. Figure 11 shows the time series of the observed and CV-predicted $PM_{2.5}$
concentrations, which demonstrates a good performance for the $PM_{2.5}$ GLM.
Figure 11 here
Table 7 shows the CV-predicted $PM_{2.5}$ concentrations. The adjusted $R^2$ values
for the five CV periods ranged from 0.710 to 0.807, which was lower than the value
(0.808) derived from the $PM_{2.5}$ model, due to the lack of input data. The observed mean
$PM_{2.5}$ concentrations were 94, 59, 44, 54, and 41 µg m$^{-3}$ for the five CV periods,
respectively. The corresponding CV-predicted mean $PM_{2.5}$ concentrations were 82, 57,
52, 65, and 47 µg m$^{-3}$, respectively. The relative error (RE) between the observed mean
$PM_{2.5}$ concentrations and the CV-predicted mean $PM_{2.5}$ concentrations ranged from −17%
to 15%, with a mean RE of −5%. The RMSE of the RE was 14.6%, reflecting the
uncertainties of the GLM method in quantitatively estimating the contribution of the
meteorological conditions to the air pollutant concentrations.
Table 7 also lists the daily RMSE for each CV period and the total RMSE. The
daily RMSE for each CV period was calculated with the daily average $PM_{2.5}$
concentrations during each CV period, and the total RMSE was calculated with the
daily average $PM_{2.5}$ concentration throughout all five CV periods combined. The daily
RMSE ranged from 19 to 53 µg m$^{-3}$, and the total RMSE was 33 µg m$^{-3}$, indicating that
the model prediction accuracy at the daily level needs to be improved. Liu et al. (2012)
used a generalized additive model (GAM) to predict $PM_{2.5}$, which had a total daily
RMSE of 23 µg m$^{-3}$. Compared with their results, the CV performance in our study was
satisfactory considering that the independent variables in our model were only based



on meteorological parameters, while the model of Liu et al. (2012) included AOD.
The relative error calculated with the CV method for GLM was −5% (Table 7),
which was smaller than the mean percentage difference (−16%) calculated based on
days with stable meteorological conditions (Table 6). Moreover, the RMSE of relative
error calculated with the CV method for GLM (Table 7) was 14.6%, which was also
smaller than the RMSE of percentage difference (18%) calculated based on days with
stable meteorological conditions (Table 6).
These indicate that the GLM reduced uncertainties of the method in
quantitatively estimating the contribution of the meteorological conditions to the
pollutant concentrations.
**3.3.2 Residual Analysis of GLM**
Table 8 shows the concentrations of air pollutants for the GLM with adjusted $R^2$
values higher than 0.6. Again, we used the $PM_{2.5}$ model as an example. Table 9 lists the
output indexes of the $PM_{2.5}$ GLM, including a model summary, analysis of variance
(ANOVA), coefficients, and other indexes. The values of R, $R^2$, and adjusted $R^2$ were
0.910, 0.828, and 0.808, respectively, indicating that the $PM_{2.5}$ model can explain 80.8%
of the variability of the daily average $PM_{2.5}$ concentrations. The model was statistically
significant according to the p-value ($<0.05$) from an F-test, and the meteorological
parameters eventually selected as the independent variables of the model were
statistically significant according to the p-values ($<0.05$) from a t-test. The
meteorological parameters eventually included in the model were lnWS, $lnWS_{max(lag)}$,
$PBL_{max}$, PREC, $\ln\Delta T_{(lag)}$, $WS_{mode}$, $WD/WS_{(lag)}$, $PBL_{min(lag)}$, $PREC_{(lag)}$, and $SLP_{min}$.
According to the collinearity statistics, all the VIF values were within 5 and tolerance
values were larger than 0.1, indicating that no serious multicollinearity existed between
the independent parameters. The Durbin–Watson value (1.910) was close to 2,
accounting for the good independence of the variance.

Figure 12 shows a residual analysis of the model. According to the residual

histogram (a), the mean value of the regression standardized residual was −0.01, with
a standard deviation of 0.955. According to the P-P graph (c), the distribution of the
observed and expected cumulative probability spread along the diagonal of $y = x$.
According to the de-trended P-P graph (d), the deviations from a normal distribution
were within ±0.05. These results indicate that the model residuals followed a normal
distribution. The scatter diagram of residuals and simulated values (b) could be applied
to test the homoscedasticity, i.e. the distribution of the regression residual did not
change over the range of values predicted by the regression. Figure S4 demonstrates
the time series of the observed pollutant and GLM-predicted pollutant concentrations,
which displayed a good correlation.

Figure 12 here

**3.3.3 Quantitative Estimates of the Contribution of Meteorological Conditions to**
**Air Pollutant Concentrations**

We applied the GLM to predict air pollutant concentrations during APEC 2014

and Parade 2015 based on meteorological parameters. The difference between the
observed and GLM-predicted concentrations was attributed to emission reduction



through the implementation of air pollution control strategies.
Table 10 lists the percentage differences between the observed and GLM-predicted
concentrations of air pollutants during APEC and Parade. The mean concentrations of
the observed and predicted $PM_{2.5}$ were 48 and 67 µg m$^{-3}$ during APEC, i.e. a 28%
difference. The mean concentrations of the observed and predicted $PM_{2.5}$ were 15 and
20 µg m$^{-3}$ during Parade, i.e. a 25% difference. These differences are attributed to the
emission reduction through the implementation of air pollution control strategies. As
described in Section 3.1, the mean concentrations of $PM_{2.5}$ decreased by 58% and 63%
during APEC and Parade, therefore, the meteorological conditions and pollution control
strategies contributed 30% and 28% to the reduction of the $PM_{2.5}$ concentration during
APEC 2014, respectively, and 38% and 25% during Parade 2015, respectively.
The emission reduction during APEC in this study is comparable to the results of
other studies where meteorological influences were considered. For example, the $PM_{2.5}$
concentration decreased by 33% under the same weather conditions during APEC in
Beijing as modelled by the Weather Research and Forecasting model and Community
Multiscale Air Quality (WRF/CMAQ) model (Wu et al., 2015). In addition, emission
control implemented in Beijing during APEC resulted in a 22% reduction in the $PM_{2.5}$
concentration, as modelled by WRF-Chem (Guo et al., 2016).
Same as $PM_{2.5}$, the differences listed in Table 10 for other pollutants show the
reduction in emission of these pollutants and/or their precursors. The differences for EC
were 37% (from 2.7 to 1.7 µg m$^{-3}$) during APEC and 33% (from 1.2 to 0.8 µg m$^{-3}$)
during Parade. In contrast, the differences for OC were 11% (from 12.6 to 11.2 µg m$^{-3}$)



during APEC and 8% (from 3.7 to 4.0 μg m$^{-3}$) during Parade. The differences for
carbonaceous components (OC + EC) were 16% (from 15.3 to 12.9 μg m$^{-3}$) during
APEC and 2% (from 4.9 to 4.8 μg m$^{-3}$) during Parade. This indicates that the emission
reduction for OC and its precursors were smaller than the reduction of EC during APEC
and Parade. A similar pattern was found for the reduction for EC and OC based on days
with stable meteorological conditions discussed in Section 3.2.2.

Table 10 also shows the differences for sulphate were 44% (from 2.7 to 3.9 μg m$^{-3}$)

during APEC and 50% (from 5.2 to 2.6 μg m$^{-3}$) during Parade. The differences for
nitrate were 44% (from 19.0 to 10.6 μg m$^{-3}$) during APEC and 56% (from 3.4 to 1.5 μg
m$^{-3}$) during Parade. The differences for ammonium were 13% (from 5.5 to 4.8 μg m$^{-3}$)
during APEC and 38% (from 2.4 to 1.5 μg m$^{-3}$) during Parade. In total, the differences
for SNA were 29% (from 27.2 to 19.3 μg m$^{-3}$) during APEC and 49% (from 11.0 to 5.6
μg m$^{-3}$) during Parade.

The concentration of sulphate is determined by primary emissions and secondary

transformation from $SO_2$; thus, the changes in sulphate concentrations may not reflect
the effectiveness of emission control strategies. One needs to also include the changes
in $SO_2$ concentrations. By adding the molar concentrations of $SO_2$ and $SO_4^{2-}$ (S = [$SO_2$]
+ [$SO_4^{2-}$]), the concentration of total S was calculated. Table 10 shows the differences
for $SO_2$ were 50% (from 6.59 to 3.32 ppb) during APEC and 2% (from 0.56 to 0.57
ppb) during Parade, while the differences for total S were 41% (from 0.322 to 0.189
μmol m$^{-3}$) during APEC and 33% (from 0.079 to 0.053 μmol m$^{-3}$) during Parade. Coal
combustion emissions is the major contributor to total S, this demonstrates the effective





control of coal combustion during both APEC 2014 and Parade 2015. The difference
for $SO_2$ during APEC was larger than that during Parade, while the difference for
sulphate during Parade was larger than that during APEC. As discussed in Section 3.1,
the mean SOR was 27% and 64% during APEC and Parade, respectively, indicating
that the $SO_2$-to-sulphate transformation rate during APEC (autumn and early winter)
was much lower than during Parade (late summer and autumn).

It is interesting to note that the difference for OC during APEC was only 11%

(Table 10) and the observed concentration of OC was even 8% higher than the GLM-
predicted concentration during Parade, indicating that the control of the OC
concentration was not as effective as the control of other $PM_{2.5}$ components during
APEC and Parade. This may be because OC can originate from both primary emission
and secondary transformation. In contrast, the control of the SNA concentration was
very effective during APEC and Parade, leading to a significant decrease of $PM_{2.5}$
during both events.

Table 10 shows $NO_x$ and other $PM_{2.5}$ components also had significant emission

reduction during APEC 2014 and Parade 2015. The differences between the observed
and GLM-predicted concentrations of $NO_x$ were 56% (from 102 to 45 ppb) during
APEC and 35% (from 20 to 13 ppb) during Parade. The differences for $Cl^-$ were 20%
(from 2.58 to 2.06 μg m$^{-3}$) during APEC and 6% (from 0.17 to 0.16 μg m$^{-3}$) during
Parade. The differences for $K^+$ were 37% (from 1.03 to 0.65 μg m$^{-3}$) during APEC and
25% (from 0.24 to 0.18 μg m$^{-3}$) during Parade. The differences for Pb, Zn, and Mn
ranged from 21% to 53% during APEC and Parade.





The concentrations of Cl⁻ have been found to be high in the fine particles produced
from coal combustion (Takuwa et al., 2006), while the concentrations of K⁺ are high in
particles derived from combustion activities, e.g. biomass burning and coal combustion.
Lead is typically considered to be a marker of emissions from coal combustion, power
stations, and metallurgical plants (Dan et al., 2004; Mukai et al., 2001; Schleicher et al.,
2011). Zinc can be produced by the action of a car braking and by tire wearing (Cyrys
et al., 2003; Sternbeck et al., 2002). Manganese mainly originates from industrial
activities. Major sources of $NO_x$ emissions include power plants, industry, and
transportation (Liu and Zhu, 2013). The differences for the concentrations of total S,
Cl⁻, K⁺, Pb, Zn, Mn, and $NO_x$, indicate that the control of anthropogenic emissions,
especially coal combustion, was very effective during APEC and Parade.
**4 Conclusions**
During the pollution control periods of APEC 2014 and Parade 2015, the
concentrations of air pollutants except ozone decreased dramatically compared with the
concentrations during non-control periods, accompanied by meteorological conditions
favourable for pollutant dispersal.
To estimate the contributions of meteorological conditions and pollution control
strategies in reducing air pollution, comparing the concentrations of air pollutants
during days with stable meteorological conditions is a useful method, but has limitation
due to high uncertainty and lack of a sufficient number of days with stable
meteorological conditions



Our study shows that, if including the nonlinear relationship between
meteorological parameters and air pollutant concentrations, GLMs based only on
meteorological parameters could provide a good explanation of the variation of
pollutant concentrations, with adjusted $R^2$ values mostly larger than 0.7. Since the
GLMs contained no parameters dependent on air pollution levels as independent
variables, they could be used to estimate the contributions of meteorological conditions
and pollution control strategies to the air pollution levels during emission control
periods.
With the GLMs method, we found meteorological conditions and pollution control
strategies played almost equally important roles in reducing air pollution in megacity
Beijing during APEC 2014 and Parade 2015, e.g. 30% and 28% to the reduction of the
$PM_{2.5}$ concentration during APEC 2014, as well as 38% and 25% during Parade 2015.
We also found that the control of the SNA concentration was more effective than
carbonaceous components. The differences between the observed and GLM-predicted
concentrations of specific pollutants ($Cl^-$, $K^+$, Pb, Zn, Mn, $NO_x$, and S) related to coal
combustion and industrial activities revealed the effective control of anthropogenic
emissions.
In the future, combining the methods of source apportionment, the contributions
of emission reductions for different sources in reducing air pollution could be estimated,
enabling further analysis of pollution control strategies.

**Data availability.** The data of stationary measurements are available upon requests.





**Author contribution.** T. Zhu and P. F. Liang designed the experiments. P. F. Liang

collected and weighed the $PM_{2.5}$ filter samples. P. F. Liang, Y. H. Fang, Y. Q. Han, and

J. X. Wang carried out the analysis of the components in $PM_{2.5}$. Y. S. Wu and M. Hu

provided the data of gaseous pollutant concentrations. Y. R. Li computed the data of

planetary boundary layer heights from GDAS and P. F. Liang developed the generalized

linear regression model. J. X. Wang managed the data. P. F. Liang analyzed the data

with contributions from all co-authors. P. F. Liang prepared the manuscript with helps

from T. Zhu.

**Acknowledgement.** This study was supported by the National Natural Science

Foundation Committee of China (41421064, 21190051), the European 7th Framework

Programme Project PURGE (265325), and the Collaborative Innovation Center for

Regional Environmental Quality.

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



**Figure Captions:**



Figure 1. Time series of atmospheric particulate matter of aerodynamic diameter ≤ 2.5
μm ($PM_{2.5}$) and the concentrations of its components, wind direction (WD), wind speed
(WS), temperature (T), and relative humidity (RH) before, during, and after (a) APEC
2014 and (b) Parade 2015. The grey-shaded areas highlight the pollution control periods
of APEC 2014 (3 November to 12 November 2014) and Parade 2015 (20 August to 3
September 2015).

Figure 2. Proportions of the measured components in $PM_{2.5}$ during (a) APEC 2014 and
(b) Parade 2015 campaigns, including organic carbon (OC), elemental carbon (EC),
$SO_4^{2-}$, $NO_3^-$, $NH_4^+$, $Cl^-$ and elements. B: before; A: after.

Figure 3. Scatter plot and correlations between organic carbon (OC: $y$-axis) and
elemental carbon (EC: $x$-axis) concentrations of $PM_{2.5}$ during the APEC 2014 and
Parade 2015 campaigns. The red symbols denote the non-control period and the black
symbols denote the pollution control period. The linear regression equations and $R^2$
values are given for these two campaigns.

Figure 4. Upper panel: time series of the proportion of sulphate, nitrate, and ammonia
(SNA) in $PM_{2.5}$ ($\rho(SNA)/PM_{2.5}$) and $PM_{2.5}$ mass concentrations (the black bar
represents $PM_{2.5}$ concentration and the red line represents $\rho(SNA)/PM_{2.5}$). Middle panel:
$SO_2$, $SO_4^{2-}$, and SOR ($[SO_4^{2-}]/([SO_2]+[SO_4^{2-}])$). Lower panel: $NO_x$, $NO_3^-$, and NOR
($[NO_3^-]/([NO_x]+[NO_3^-])$). Data collected during the (a) APEC 2014 and (b) Parade





2015 campaigns. The hollow bars represent gaseous pollutants (red for $SO_2$, blue for
$NO_x$), and solid bars represent secondary inorganic ions (red for sulphate, blue for
nitrate).

Figure 5. Wind rose plots based on frequencies of half-hourly data before APEC
(BAPEC), during APEC, and after APEC (AAPEC) on the left, and before Parade
(BParade), during Parade, and after Parade (AParade) on the right.

Figure 6. Time series of daily $PM_{2.5}$ concentrations and planetary boundary layer (PBL)
heights during the (a) APEC and (b) Parade campaigns. The black line represents $PM_{2.5}$
concentrations and the red line represents PBL heights. The grey-shaded areas highlight
the pollution control periods of APEC 2014 (3 November to 12 November 2014) and
Parade 2015 (20 August to 3 September 2015).

Figure 7. Scatter plot showing the correlation between daily $PM_{2.5}$ concentrations ($y$-
axis) and (a) daily PBL heights ($x$-axis) and (b) daily WSs ($x$-axis) during the APEC
2014 sampling period (October to December 2014) and Parade 2015 sampling period
(August to December 2015). The red and black scattered points represent different
distribution areas. The piecewise function regression equations and the corresponding
values of PBL height and WS according to the intersections are given.

Figure 8. The black bars represent the percentage reductions calculated by comparing
the decreased average concentrations during APEC to the average concentrations
before APEC. The red bars represent the percentage reductions calculated by comparing



the decreased average concentrations during APEC to the average concentrations
before APEC based only on the days with stable meteorological conditions. The
whiskers represent the standard deviations of the percentage reductions.

Figure 9. Variations of air pollutant concentrations during days with stable
meteorological conditions during the APEC 2014 campaign, including $PM_{2.5}$, sulphate,
nitrate, and ammonium (SNA), organic carbon (OC), elemental carbon (EC), $Cl^-$, $K^+$,
elements (Pb, Zn, Ni, Mn, and Ca), and gaseous pollutants ($SO_2$, NO, $NO_x$, and $O_3$).
The red points represent mean values. The black cross bars are median values. The
black box denotes the $25^{th}$ and $75^{th}$ percentiles. The whiskers represent the maximum
and minimum, respectively. B: before; A: after.

Figure 10. Scatter plot and correlations between GLM-predicted (*y*-axis) and observed
(*x*-axis) concentrations of pollutants transformed to a natural log. The linear regression
equations and $R^2$ values are given.

Figure 11. Time series of the observed and cross-validation (CV) predicted $PM_{2.5}$
concentrations during five CV periods. The black line represents the observed $PM_{2.5}$
concentration and the red line represents the CV-predicted $PM_{2.5}$ concentration.

Figure 12. Residual analysis of the model. (a) Histogram of the regression standardized
residual. (b) Scatter plot between the regression standardized predicted value and
regression standardized residual. (c) Normal P-P plot of the regression standardized





residual between the observed cumulative probability and expected cumulative
probability. (d) De-trended normal P-P plot of the standardized residual of observed
cumulative probability.






Table 1. Air pollution control strategies during APEC 2014 and Parade 2015.

| Periods | Control measures | Detail of measures |
|---|---|---|
| **APEC 2014** (3 to 12 November 2014) and **Parade 2015** (20 August to 3 September 2015) | Traffic control | The odd/even plate number rule for traffic control in Beijing, Tianjin, Hebei and Shandong; 70% (APEC 2014)/80% (Parade 2015) of official vehicle and "yellow label vehicles" were banned from Beijing's roads; Trucks limited to run inside the 6th Ring Road between 6 AM to 24 PM. |
| | Industrial emission control | More than 10,000 factories production limited or halted in Beijing and Hebei, Tianjin, Shandong, Shanxi and Inner Mongolia which surround Beijing city. |
| | Dust pollution control | Dust emission factories and outdoor constructions shut down or limited in Beijing and near area; Enhancing road cleaning and spray and aspirating in Beijing. |
| | Coal-fired control | State-owned enterprise productions enhancing limited and 40% coal-fired boilers shut down in Beijing; more special pollutant emission factory limited around Beijing. |







Table 2. Summary of statistical models applied to predict air pollutant concentrations
with meteorological parameters.

| Dependent variables | Independent variables | $R^2$ | Methods* | Applications |
|---|---|---|---|---|
| $PM_{2.5}$ | meteorological parameters (T/RH/PBL/WS/cloud fraction), AOT | 0.47 | MLR | Gupta and Christopher, 2009 |
| $PM_{10}$ | meteorological parameters (T/WD/RH/PBL/WS), AOD | 0.21/0.30 (MODIS/MISR) | MLR | Sotoudeheian and Arhami, 2014 |
| $PM_{10}$ | meteorological parameters (RH/WS/T), AOD | 0.49-0.88 (spatial-temporal variability) | MLR | Chitranshi et al., 2015 |
| $PM_{2.5}$ | meteorological parameters (T/RH/PREC), AOT | 0.60/0.58 (MOD/MYD) | MLR | Nguyen et al., 2015 |
| $\ln(PM_{2.5})$, $\ln(PM_{2.5-10})$ | meteorological parameters $(\ln(PREC)/\ln(RH)/\ln(WS)/\ln(SUN)/\ln(T))$, atmospheric turbulence parameters $(\ln(\Delta u/\Delta z)/\ln(\Delta\theta/\Delta z))$ | 0.60-0.74 | GLM | Hien et al., 2002 |
| $\ln(PM_{2.5})$ | meteorological parameters $(T/WD/\ln(WS)/\ln(PBL))$, $\ln(AOT)$, categorical parameters | 0.51/0.62 (MODIS/MISR) | GLM | Liu et al., 2007 |
| $\log(PM_{2.5})$, $\log(BC)$ | meteorological parameters (T/wind index), traffic-related parameters | 0.62/0.42 ($PM_{2.5}$/BC) | GLM | Richmond-Bryant et al., 2009 |
| $\ln(PM_{2.5})$ | meteorological parameters $(\ln(PBL)/GEO-4 RH/\ln(surface RH)/T)$, $\ln(AOD)$ | 0.65 | GLM | Tian and Chen, 2010 |
| $\ln(PM_{10})$ | meteorological parameters $(T/WD/RH/\ln(PBL)/\ln(WS))$, $\ln(AOD)$ | 0.18/0.38 (MODIS/MISR) | GLM | Sotoudeheian and Arhami, 2014 |
| $\ln(PM_{2.5})$ | meteorological parameters $(\ln(PBL)/RH/Vis/\ln(T)/\ln(WS))$, $\ln(AOD)$ | 0.67/0.72 (MODIS/MISR) | GLM | You et al., 2015 |
| $\ln(PM_{2.5})$ | meteorological parameters (WS/WD/T/RH/pressure), optical properties (absorption/scattering/attenuation co-efficient) | 0.54/0.31/0.32/0.88 (winter/pre-monsoon/monsoon/post-monsoon) | GLM | Raman and Kumar, 2016 |
| $PM_{10}$, $PM_{2.5}$ | smooth non-parametric functions of spatial/temporal variates | 0.58 | GAM | Barmpadimos et al., 2012 |
| $PM_{2.5}$, $PM_{10}$, $PM_{2.5-10}$ | smooth non-parametric functions of spatial/temporal variates | 0.77/0.58/0.46-0.52 ($PM_{2.5}$/$PM_{10}$/$PM_{2.5-10}$) | GAM | Yanosky et al., 2014 |
| $PM_{10}$ | meteorological parameters $(WS/T_{min}/T_{max})$, previous day $PM_{10}$ | 0.78 | ANN | Diaz-Robles et al., 2008 |
| $PM_{2.5}$ | meteorological parameters (WS/RH/PBL/WS*PBL), AOD, spatial | 0.89 | LUR | Chudnovsky et al., 2014 |





| | explanatory variables | | | |
|---|---|---|---|---|
| $PM_{10}$, $NO_2$ | meteorological parameters (T/RH/WS/air pressure/cloud cover/percentage of haze/mist/rain/sun), spatial explanatory variables | 0.45/0.43 ($PM_{10}$/$NO_2$) | LUR | Liu et al., 2015 |

*MLR: multiple linear regression model, GLM: generalized linear regression model, GAM: generalized additive model, ANN: artificial neural networks, LUR: land use regression model.







Table 3. Meteorological parameters used in the GLM. The calculation of each
meteorological parameter is based on the sample duration of 23.5 h (09:30–09:00 LT
the next day).

| Parameters | Abbreviations | Description |
|---|---|---|
| Wind direction value* | WD | The average of wind direction values. |
| | $WD_{sum}$ | The sum of wind direction values. |
| | $WD_{mode}$ | The mode of wind direction values. |
| Wind speed (m s$^{-1}$) | WS | The average of wind speed. |
| | $WS_{mode}$ | The mode of wind speed. |
| | $WS_{max}$ | The maximum of wind speed. |
| Temperature (℃) | T | The average of temperature. |
| | $T_{max}$ | The maximum of temperature. |
| | $T_{min}$ | The minimum of temperature. |
| | $\Delta T$ | The difference of temperature. |
| Sea level pressure (hPa) | SLP | The average of sea level pressure. |
| | $SLP_{max}$ | The maximum of sea level pressure. |
| | $SLP_{min}$ | The minimum of sea level pressure. |
| Relative humidity (%) | RH | The average of relative humidity. |
| | $RH_{max}$ | The maximum of relative humidity. |
| Precipitation (mm) | PREC | The accumulation of precipitation. |
| Wind index | WD/WS | The average of wind direction value/wind speed. |
| | $WD/WS_{sum}$ | The sum of wind direction value/wind speed. |
| Planetary boundary layer height (m) | PBL | The average of 3-h planetary boundary layer height. |
| | $PBL_{min}$ | The minimum of 3-h planetary boundary layer height. |
| | $PBL_{max}$ | The maximum of 3-h planetary boundary layer height. |

* Since the degree data of wind direction cannot be applied directly, the values of wind
directions are donated such that value = 1, 2, 3 for north, south, and "calm and
variable", respectively.






Table 4. Statistical summary showing the mean concentrations and standard deviations
of PM$_{2.5}$, gaseous pollutants, and PM$_{2.5}$ components. B: before; A: after.

| Pollutants | Units | BAPEC | APEC | AAPEC | BParade | Parade | AParade |
|---|---|---|---|---|---|---|---|
| PM$_{2.5}$ | | 113±62 | 48±35 | 97±84 | 41±14 | 15±6 | 39±28 |
| OC | | 15.3±8.7 | 11.2±7.2 | 21.3±15.5 | 7.4±1.9 | 4.0±1.0 | 6.3±3.1 |
| EC | | 2.7±1.4 | 1.7±1.0 | 3.5±1.8 | 1.6±0.3 | 0.8±0.1 | 2.0±1.0 |
| SO$_4^{2-}$ | | 12.6±9.1 | 3.9±3.0 | 9.6±12.4 | 10.6±6.2 | 2.6±1.3 | 7.9±7.3 |
| NO$_3^-$ | | 29.4±21.4 | 10.6±11.0 | 16.3±19.4 | 5.0±3.9 | 1.5±1.5 | 6.4±6.2 |
| NH$_4^+$ | | 15.0±10.6 | 4.8±4.2 | 10.3±11.9 | 5.2±2.6 | 1.5±1.0 | 5.4±5.4 |
| Cl$^-$ | | 3.19±1.61 | 2.06±2.11 | 6.59±6.67 | 0.20±0.16 | 0.16±0.12 | 0.53±0.24 |
| Na$^+$ | μg m$^{-3}$ | 0.50±0.26 | 0.26±0.15 | 0.57±0.46 | 0.16±0.09 | 0.10±0.05 | 0.16±0.08 |
| K$^+$ | | 1.20±0.63 | 0.65±0.51 | 1.52±1.43 | 0.30±0.13 | 0.18±0.08 | 0.38±0.20 |
| Mg$^{2+}$ | | 0.07±0.03 | 0.09±0.02 | 0.13±0.07 | 0.01±0.01 | 0.01±0.00 | 0.02±0.01 |
| Ca$^{2+}$ | | 0.52±0.34 | 0.28±0.19 | 0.53±0.40 | 0.14±0.07 | 0.10±0.04 | 0.17±0.05 |
| SO$_2$ | | 11.3±5.0 | 9.5±6.8 | 34.8±15.3 | 2.7±1.6 | 1.6±1.4 | 5.9±5.2 |
| NO | | 54.2±30.5 | 21.9±13.8 | 112.3±63.2 | 3.2±2.1 | 1.2±0.9 | 9.3±7.5 |
| NO$_x$ | | 151±62 | 81±46 | 220±107 | 57±11 | 26±13 | 63±24 |
| O$_3$ | | 23±16 | 38±19 | 17±14 | 116±33 | 79±22 | 74±27 |
| Ca | | 582±431 | 591±335 | 1536±579 | 202±64 | 108±36 | 188±130 |
| Co | | 0.48±0.21 | 0.34±0.18 | 0.90±0.52 | 0.21±0.08 | 0.05±0.02 | 0.16±0.10 |
| Ni | | 3.20±1.56 | 5.07±7.42 | 5.17±2.50 | 1.75±1.16 | 0.63±0.72 | 1.16±0.67 |
| Cu | | 35.7±16.2 | 19.1±12.6 | 43.3±31.2 | 12.4±5.1 | 3.7±1.3 | 9.6±6.5 |
| Zn | | 320±146 | 128±120 | 315±310 | 97±46 | 20±9 | 71±54 |
| Se | | 6.45±3.46 | 3.76±3.84 | 5.22±6.56 | 7.06±3.41 | 3.19±2.76 | 3.17±2.76 |
| Mo | | 2.20±1.12 | 1.63±1.14 | 2.85±2.67 | 0.62±0.41 | 0.16±0.14 | 0.53±0.46 |
| Cd | | 3.86±2.53 | 1.41±1.25 | 3.11±2.52 | 2.35±5.72 | 0.22±0.17 | 0.71±0.74 |
| Tl | ng m$^{-3}$ | 1.87±0.90 | 0.87±1.01 | 2.03±1.96 | 0.50±0.31 | 0.05±0.06 | 0.33±0.39 |
| Pb | | 121±59 | 55±52 | 104±81 | 36±19 | 9±6 | 29±26 |
| Th | | 0.09±0.05 | 0.06±0.03 | 0.09±0.06 | 0.02±0.01 | 0.01±0.01 | 0.01±0.01 |
| U | | 0.06±0.02 | 0.05±0.03 | 0.09±0.06 | 0.02±0.01 | 0.00±0.00 | 0.01±0.02 |
| Na | | 529±261 | 355±209 | 907±632 | 182±71 | 96±39 | 181±96 |
| Mg | | 153±94 | 105±47 | 236±143 | 43±13 | 15±8 | 24±15 |
| Al | | 516±324 | 338±154 | 588±406 | 141±82 | 130±60 | 136±93 |
| Mn | | 55.5±23.3 | 34.5±24.1 | 61.6±52.4 | 17.3±6.4 | 3.6±1.8 | 14.8±9.2 |
| Fe | | 755±314 | 573±336 | 883±538 | 269±71 | 98±28 | 234±139 |
| Ba | | 16.3±8.0 | 11.0±8.4 | 13.8±8.1 | 4.7±1.6 | 1.9±0.6 | 4.1±2.3 |







Table 5. Statistical summary showing the meteorological conditions (WS and PBL
height), and the concentrations of pollutants on the days with stable meteorological
conditions during the APEC campaign. B: before; A: after.

| | | WS | PBL | $PM_{2.5}$ | OC | EC | $SO_4^{2-}$ | $NO_3^-$ | $NH_4^+$ | $SO_2$ | NO | $NO_x$ | $O_3$ |
|---|---|---|---|---|---|---|---|---|---|---|---|---|---|
| | | (m s$^{-1}$) | (m) | | | | ($\mu$g m$^{-3}$) | | | | | | |
| BAPEC | 10/18 | 1.27 | 145 | 153 | - | - | 14.3 | 43.0 | 19.4 | - | 73 | 240 | 45.2 |
| | 10/22 | 1.46 | 233 | 67 | 10.9 | 3.6 | 7.4 | 14.0 | 9.3 | 12.1 | 59 | 153 | 4.4 |
| | 10/23 | 1.46 | 188 | 108 | 18.0 | 2.6 | 11.3 | 23.7 | 13.2 | 11.5 | 91 | 200 | 6.6 |
| | 10/24 | 1.52 | 171 | 177 | 20.5 | 3.0 | 24.6 | 54.2 | 28.3 | 12.0 | 73 | 205 | 20.4 |
| | 10/28 | 1.71 | 232 | 87 | 15.9 | 3.8 | 7.0 | 17.8 | 7.9 | 13.0 | 69 | 165 | 14.2 |
| | 10/29 | 1.10 | 193 | 132 | 23.6 | 5.3 | 10.6 | 35.2 | 14.0 | 15.6 | 99 | 229 | 10.0 |
| | 10/30 | 1.00 | 160 | 170 | 26.0 | 4.0 | 18.5 | 56.0 | 25.7 | 18.4 | 77 | 195 | 7.5 |
| | 10/31 | 1.50 | 209 | 138 | 17.0 | 3.9 | 13.5 | 29.3 | 16.1 | 8.1 | 79 | 183 | 4.6 |
| | **Mean** | **1.38** | **191** | **129** | **18.8** | **3.7** | **13.4** | **34.2** | **16.7** | **12.9** | **77** | **196** | **14.1** |
| APEC | 11/3 | 1.98 | 211 | 39 | 11.1 | 1.8 | 1.8 | 4.9 | 2.6 | 5.8 | 19 | 84 | 36.5 |
| | 11/4 | 1.85 | 163 | 116 | 22.7 | 2.9 | 9.6 | 33.1 | 13.2 | 26.0 | 31 | 144 | 20.5 |
| | 11/7 | 1.63 | 264 | 59 | 12.5 | 2.8 | 4.3 | 10.9 | 6.1 | 13.6 | 30 | 101 | 15.0 |
| | 11/8 | 2.00 | 196 | 76 | 17.3 | 2.4 | 7.3 | 21.1 | 8.8 | 11.8 | 26 | 101 | 33.6 |
| | 11/9 | 1.79 | 154 | 66 | 17.6 | 2.5 | 4.9 | 14.0 | 6.3 | 9.2 | 44 | 125 | 27.7 |
| | 11/10 | 2.13 | 177 | 61 | 14.3 | 1.8 | 5.6 | 17.9 | 7.5 | 10.2 | 31 | 115 | 29.2 |
| | **Mean** | **1.90** | **194** | **70** | **15.9** | **2.4** | **5.6** | **17.0** | **7.4** | **12.7** | **30** | **112** | **27.1** |
| AAPEC | 11/14 | 1.58 | 169 | 61 | 15.2 | 4.5 | 3.4 | 6.4 | 4.1 | 30.3 | 87 | 171 | 14.7 |
| | 11/15 | 1.38 | 173 | 118 | 24.2 | 6.6 | 7.2 | 19.6 | 10.6 | 52.0 | 148 | 276 | 5.0 |
| | 11/17 | 2.48 | 252 | 57 | 14.5 | 3.8 | 2.8 | 4.0 | 3.7 | 30.4 | 125 | 206 | 25.0 |
| | 11/18 | 1.44 | 106 | 101 | 27.1 | 3.8 | 6.3 | 14.3 | 8.3 | 54.5 | 162 | 285 | 6.2 |
| | 11/19 | 1.23 | 121 | 267 | 53.2 | 5.0 | 38.2 | 55.6 | 35.2 | 54.6 | 190 | 369 | 1.0 |
| | 11/20 | 1.94 | 120 | 220 | 41.6 | 3.7 | 26.2 | 46.9 | 28.8 | 38.8 | 200 | 383 | 2.9 |
| | 11/22 | 1.96 | 178 | 58 | 14.0 | 3.3 | 3.3 | 5.4 | 4.5 | 32.2 | 89 | 183 | 26.7 |
| | **Mean** | **1.72** | **160** | **126** | **27.1** | **4.4** | **12.5** | **21.7** | **13.6** | **41.8** | **143** | **268** | **11.6** |




Table 6. The percentage differences (PD) for the $PM_{2.5}$ concentrations of four periods
(P1, P2, P3, and P4) that were randomly selected from within the non-control stable
days of the APEC 2014 and Parade 2015 campaigns.

| Periods | Mean values ($\mu g\ m^{-3}$) | SD ($\mu g\ m^{-3}$) | Total SD ($\mu g\ m^{-3}$) | Percentage differences (PD)* | | | | Mean PD | RMSE of PD |
|---|---|---|---|---|---|---|---|---|---|
| | | | | P1 | P2 | P3 | P4 | | |
| P1 | 120 | 97 | | - | - | - | - | | |
| P2 | 101 | 58 | 59 | -16% | - | - | - | -16% | 18% |
| P3 | 96 | 40 | | -20% | -5% | - | - | | |
| P4 | 87 | 23 | | -28% | -14% | -9% | - | | |

* Percentage difference (PD) = (Mean value of $P_{n+1}$ - Mean value of $P_n$)/Mean value of $P_n \times 100\%$.







Table 7. The cross-validation (CV) performance of the PM$_{2.5}$ GLM.

| Periods | Adjusted R$^2$ | Observed mean values (µg m$^{-3}$) | Predicted mean values (µg m$^{-3}$) | Daily RMSE (µg m$^{-3}$) | Total RMSE (µg m$^{-3}$) | Relative errors (RE)* | Mean RE | RMSE of RE |
|---|---|---|---|---|---|---|---|---|
| CV1 | 0.748 | 94 | 82 | 53 | | 15% | | |
| CV2 | 0.798 | 59 | 57 | 20 | | 4% | | |
| CV3 | 0.783 | 44 | 52 | 19 | 33 | -15% | -5% | 14.6% |
| CV4 | 0.710 | 54 | 65 | 27 | | -17% | | |
| CV5 | 0.807 | 41 | 47 | 30 | | -13% | | |

*Relative error (RE) = (Predicted mean value - Observed mean value)/Predicted mean value × 100%.






Table 8. The concentrations of air pollutants for the GLM with adjusted $R^2$ values higher
than 0.6.

| Pollutants | Model descriptions | Adjusted $R^2$ |
|---|---|---|
| $PM_{2.5}$ | $\ln(PM_{2.5})=-0.48\ln WS-0.43\ln WS_{max(lag)}-0.00076PBL_{max}-0.11PREC+0.25\ln\Delta T_{(lag)}-0.14WS_{mode}+0.48WD/WS_{(lag)}+0.0043PBL_{min(lag)}-0.025PREC_{(lag)}-0.015SLP_{min}+19.51$ | 0.808 |
| EC | $\ln(EC)=0.60\ln WD/WS_{sum}-0.59\ln PBL-0.017PREC_{(lag)}+0.22\ln\Delta T-0.50\ln WS_{(lag)}+0.25\ln PBL_{max(lag)}-0.17$ | 0.780 |
| OC | $\ln(OC)=-0.44\ln WS+0.47WD/WS_{(lag)}-0.67\ln PBL-0.020PREC_{(lag)}+0.67\ln WD+0.17\ln\Delta T-0.65\ln RH_{max(lag)}+7.84$ | 0.751 |
| $SO_4^{2-}$ | $\ln(SO_4^{2-})=-0.99\ln WS_{(lag)}+0.066T_{min}-0.040PREC_{(lag)}-1.20\ln PBL+0.0011PBL_{(lag)}+0.019RH-0.12PREC+0.087WS_{max}+6.68$ | 0.795 |
| $NO_3^-$ | $\ln(NO_3^-)=-1.90\ln PBL-0.96\ln WS_{(lag)}+0.88WD+0.0045PBL_{min}-0.20PREC+0.12WS_{max}+1.57\ln RH+0.60\ln\Delta T_{(lag)}-1.22\ln RH_{max(lag)}-0.047\Delta T+9.32$ | 0.833 |
| $NH_4^+$ | $\ln(NH_4^+)=0.040RH-1.27\ln WS_{(lag)}-1.03\ln RH_{(lag)}-0.00075PBL_{max}-0.16PREC+0.33\ln\Delta T_{(lag)}+4.28$ | 0.813 |
| $Cl^-$ | $\ln(Cl^-)=-1.12\ln PBL-0.072T_{(lag)}+1.60\ln WD-2.32\ln RH_{max(lag)}+0.53\ln WD/WS_{sum(lag)}+14.69$ | 0.737 |
| $K^+$ | $\ln(K^+)=-0.75\ln PBL-0.66\ln WS_{(lag)}-0.020RH_{(lag)}+0.0056PBL_{min}-0.20WS_{mode}+0.33\ln\Delta T_{(lag)}-0.47\ln PBL_{max(lag)}-0.087PREC+0.66\ln RH+5.46$ | 0.717 |
| Pb | $\ln(Pb)=-0.61\ln WS-0.67\ln WS_{max(lag)}+0.36\ln\Delta T_{(lag)}-0.00062PBL_{max}-0.19WS_{mode}-0.030PREC_{(lag)}+5.39$ | 0.721 |
| Zn | $\ln(Zn)=-0.81\ln WS-0.41\ln WS_{max(lag)}-0.0016PBL-0.36\ln WS_{mode(lag)}+6.56$ | 0.627 |
| Mn | $\ln(Mn)=0.80WD/WS-0.98\ln PBL-0.043PREC_{(lag)}+0.57WD/WS_{(lag)}-0.017RH-0.023SLP+0.0030PBL_{min(lag)}+31.04$ | 0.656 |
| $SO_2$ | $\ln(SO_2)=-1.32\ln PBL-0.071PREC_{(lag)}-0.047PREC+0.29WD_{mode(lag)}-0.026RH-0.47\ln WS_{(lag)}+14.12\ln SLP_{max}-87.56$ | 0.803 |
| $NO_x$ | $\ln(NO_x)=0.014WD/WS_{sum}-0.030T_{min}+0.27\ln\Delta T-0.44\ln PBL-0.015PREC-0.012PREC_{(lag)}+5.30$ | 0.772 |






Table 9. The output indexes of the PM$_{2.5}$ GLM, including a model summary, analysis
of variance (ANOVA), coefficients, and other indexes.

| Model Summary and ANOVA | | | | | | |
|---|---|---|---|---|---|---|
| R | R$^2$ | Adjusted R$^2$ | Std. Error of the Estimate | Durbin-Watson | F | Sig.* |
| 0.910 | 0.828 | 0.808 | 0.411 | 1.910 | 41.763 | 0.000 |
| Coefficients | | | | | | |
| Model | Unstandardized Coefficients | | t | Sig.* | Collinearity Statistics | |
| | B | Std. Error | | | Tolerance | VIF |
| (Constant) | 19.512 | 6.871 | 2.840 | 0.006 | | |
| lnWS | -0.483 | 0.162 | -2.971 | 0.004 | 0.313 | 3.194 |
| lnWS$_{max(lag)}$ | -0.431 | 0.153 | -2.818 | 0.006 | 0.300 | 3.331 |
| PBL$_{max}$ | -0.001 | 0.000 | -6.747 | 0.000 | 0.395 | 2.534 |
| PREC | -0.110 | 0.029 | -3.735 | 0.000 | 0.618 | 1.618 |
| ln$\triangle$T$_{(lag)}$ | 0.247 | 0.083 | 2.975 | 0.004 | 0.662 | 1.512 |
| WS$_{mode}$ | -0.135 | 0.050 | -2.726 | 0.008 | 0.493 | 2.027 |
| WD/WS$_{(lag)}$ | 0.476 | 0.148 | 3.222 | 0.002 | 0.353 | 2.829 |
| PBL$_{min(lag)}$ | 0.004 | 0.001 | 3.510 | 0.001 | 0.407 | 2.459 |
| PREC$_{(lag)}$ | -0.025 | 0.009 | -2.796 | 0.006 | 0.707 | 1.415 |
| SLP$_{min}$ | -0.015 | 0.007 | -2.176 | 0.032 | 0.707 | 1.414 |

*The significance level is 0.05.




Table 10. The percentage differences between the observed and GLM-predicted
concentrations of the air pollutants during APEC and Parade.

| Pollutants | Units | During APEC | | | During Parade | | |
|---|---|---|---|---|---|---|---|
| | | Observed | Predicted | Percentage differences[1] | Observed | Predicted | Percentage differences[1] |
| $PM_{2.5}$ | | 48 | 67 | 28% | 15 | 20 | 25% |
| OC | | 11.2 | 12.6 | 11% | 4.0 | 3.7 | -8% |
| EC | | 1.7 | 2.7 | 37% | 0.8 | 1.2 | 33% |
| $SO_4^{2-}$ | $\mu g\ m^{-3}$ | 3.9 | 2.7 | -44% | 2.6 | 5.2 | 50% |
| $NO_3^-$ | | 10.6 | 19.0 | 44% | 1.5 | 3.4 | 56% |
| $NH_4^+$ | | 4.8 | 5.5 | 13% | 1.5 | 2.4 | 38% |
| $Cl^-$ | | 2.06 | 2.58 | 20% | 0.16 | 0.17 | 6% |
| $K^+$ | | 0.65 | 1.03 | 37% | 0.18 | 0.24 | 25% |
| Pb | | 55 | 70 | 21% | 9 | 17 | 47% |
| Zn | $ng\ m^{-3}$ | 128 | 171 | 25% | 20 | 41 | 51% |
| Mn | | 34.5 | 51.5 | 33% | 3.6 | 7.6 | 53% |
| $SO_2$ | ppb | 3.32 | 6.59 | 50% | 0.57 | 0.56 | -2% |
| $NO_x$ | | 45 | 102 | 56% | 13 | 20 | 35% |
| OC+EC | $\mu g\ m^{-3}$ | 12.9 | 15.3 | 16% | 4.8 | 4.9 | 2% |
| SNA | $\mu g\ m^{-3}$ | 19.3 | 27.2 | 29% | 5.6 | 11.0 | 49% |
| total $S^2$ | $\mu mol\ m^{-3}$ | 0.189 | 0.322 | 41% | 0.053 | 0.079 | 33% |

[1]Percentage difference = (Predicted - Observed)/Predicted $\times$ 100%.

[2]total S = $[SO_2]$ + $[SO_4^{2-}]$.






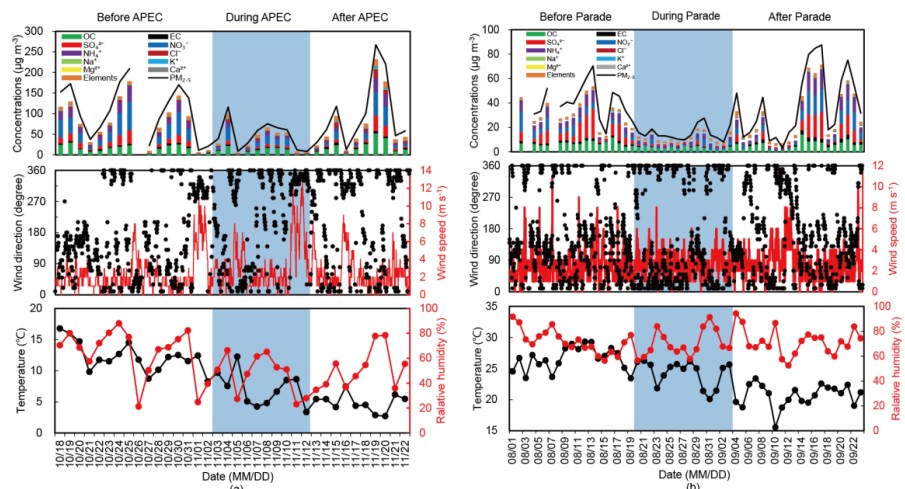


Figure 1. Time series of atmospheric particulate matter of aerodynamic diameter ≤ 2.5

μm (PM$_{2.5}$) and the concentrations of its components, wind direction (WD), wind speed

(WS), temperature (T), and relative humidity (RH) before, during, and after (a) APEC

2014 and (b) Parade 2015. The grey-shaded areas highlight the pollution control periods

of APEC 2014 (3 November to 12 November 2014) and Parade 2015 (20 August to 3

September 2015).






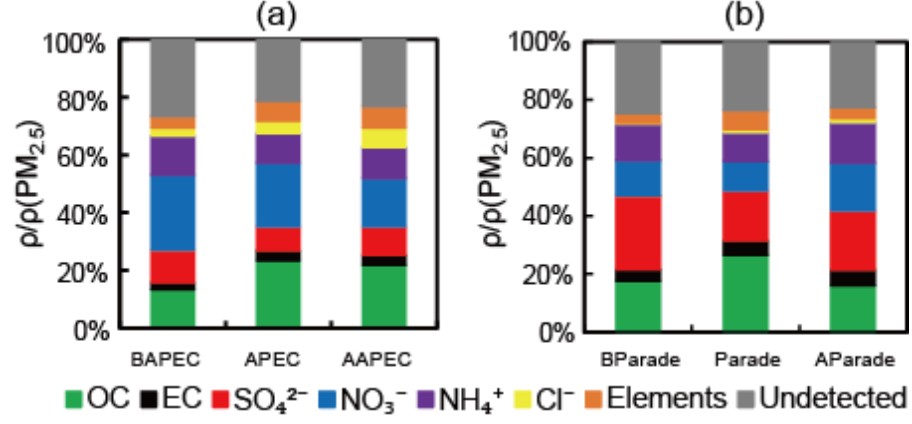


Figure 2. Proportions of the measured components in PM$_{2.5}$ during (a) APEC 2014 and
(b) Parade 2015 campaigns, including organic carbon (OC), elemental carbon (EC),
SO$_4^{2-}$, NO$_3^-$, NH$_4^+$, Cl$^-$ and elements. B: before; A: after.



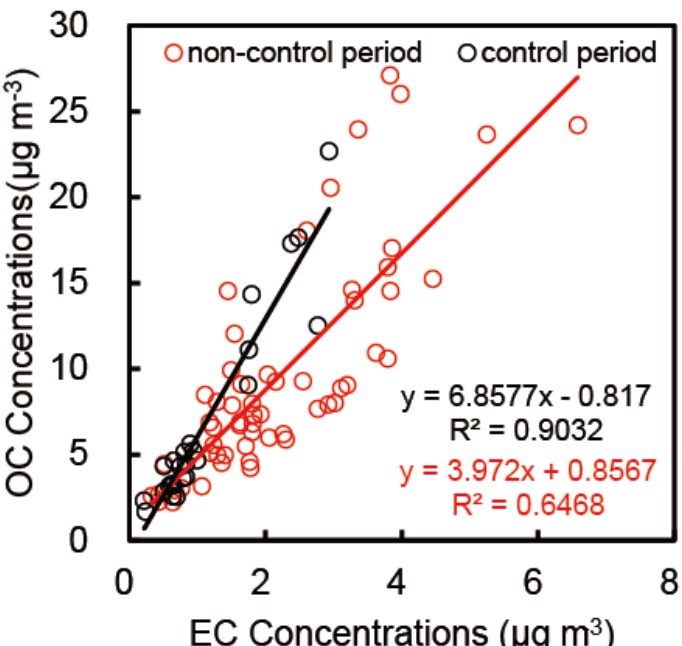


Figure 3. Scatter plot and correlations between organic carbon (OC: *y*-axis) and
elemental carbon (EC: *x*-axis) concentrations of $PM_{2.5}$ during the APEC 2014 and
Parade 2015 campaigns. The red symbols denote the non-control period and the black
symbols denote the pollution control period. The linear regression equations and $R^2$
values are given for these two campaigns.





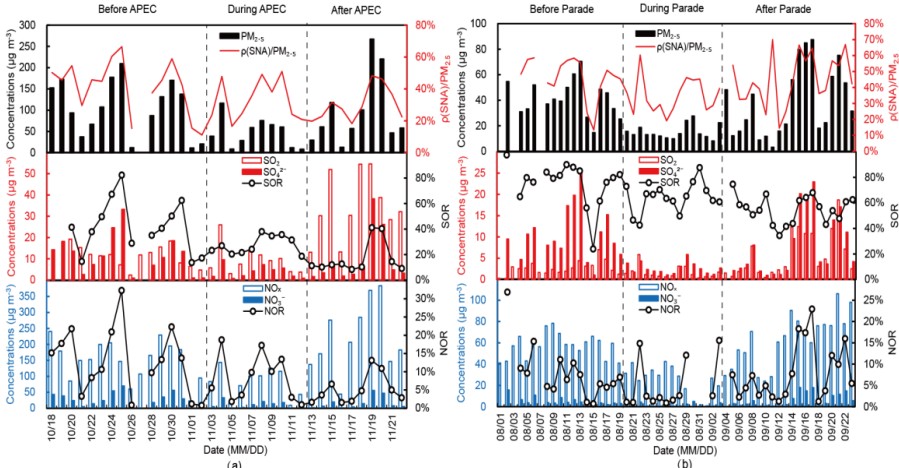


Figure 4. Upper panel: time series of the proportion of sulphate, nitrate, and ammonia
(SNA) in PM$_{2.5}$ (ρ(SNA)/PM$_{2.5}$) and PM$_{2.5}$ mass concentrations (the black bar
represents PM$_{2.5}$ concentration and the red line represents ρ(SNA)/PM$_{2.5}$). Middle panel:
SO$_2$, SO$_4^{2-}$, and SOR ([SO$_4^{2-}$]/([SO$_2$]+[SO$_4^{2-}$])). Lower panel: NO$_x$, NO$_3^-$, and NOR
([NO$_3^-$]/([NO$_x$]+[NO$_3^-$])). Data collected during the (a) APEC 2014 and (b) Parade
2015 campaigns. The hollow bars represent gaseous pollutants (red for SO$_2$, blue for
NO$_x$), and solid bars represent secondary inorganic ions (red for sulphate, blue for
nitrate).





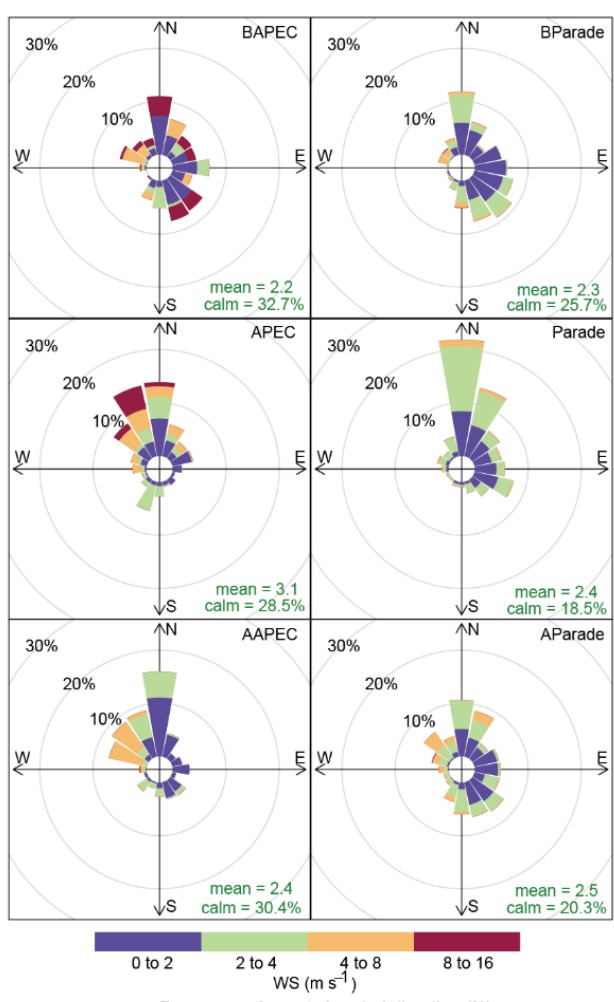


Figure 5. Wind rose plots based on frequencies of half-hourly data before APEC

(BAPEC), during APEC, and after APEC (AAPEC) on the left, and before Parade

(BParade), during Parade, and after Parade (AParade) on the right.






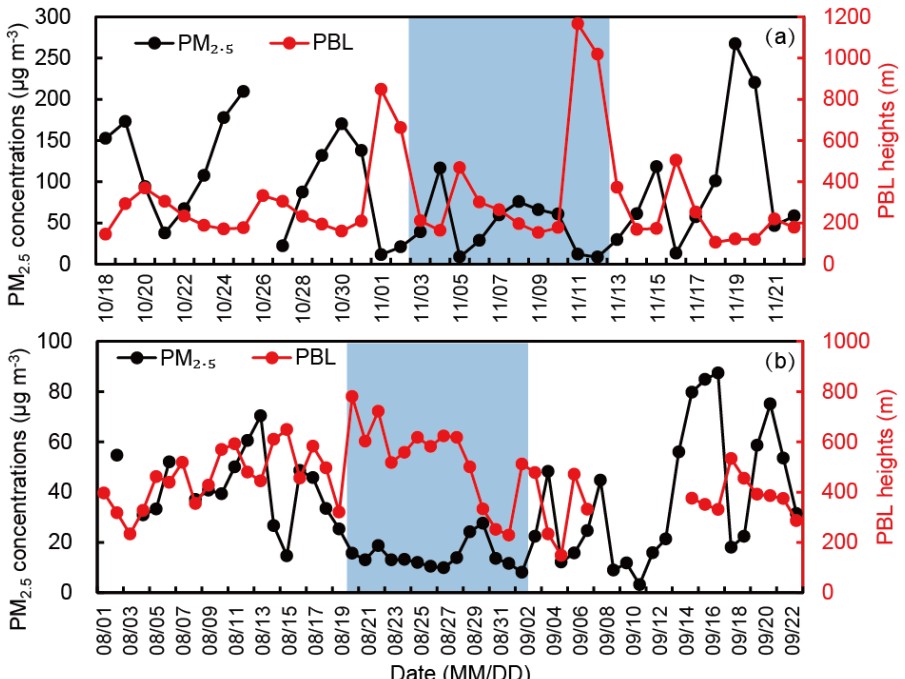


Figure 6. Time series of daily PM₂.₅ concentrations and planetary boundary layer (PBL)
heights during the (a) APEC and (b) Parade campaigns. The black line represents PM₂.₅
concentrations and the red line represents PBL heights. The grey-shaded areas highlight
the pollution control periods of APEC 2014 (3 November to 12 November 2014) and
Parade 2015 (20 August to 3 September 2015).





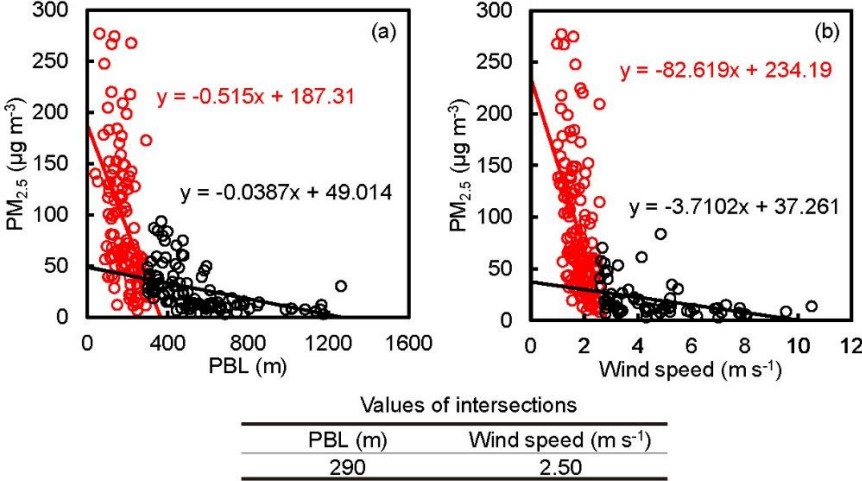


Figure 7. Scatter plot showing the correlation between daily $PM_{2.5}$ concentrations ($y$-
axis) and (a) daily PBL heights ($x$-axis) and (b) daily WSs ($x$-axis) during the APEC
2014 sampling period (October to December 2014) and Parade 2015 sampling period
(August to December 2015). The red and black scattered points represent different
distribution areas. The piecewise function regression equations and the corresponding
values of PBL height and WS according to the intersections are given.





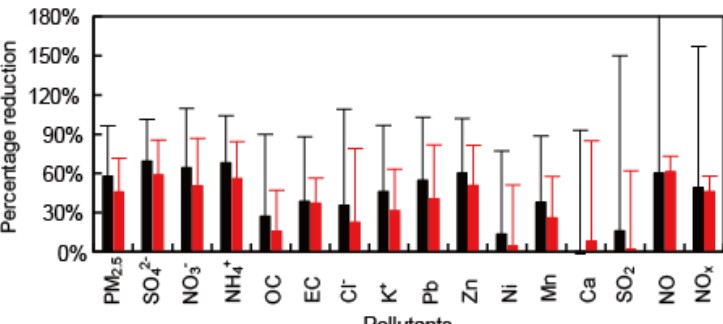


Figure 8. The black bars represent the percentage reductions calculated by comparing
the decreased average concentrations during APEC to the average concentrations
before APEC. The red bars represent the percentage reductions calculated by comparing
the decreased average concentrations during APEC to the average concentrations
before APEC based only on the days with stable meteorological conditions. The
whiskers represent the standard deviations of the percentage reductions.



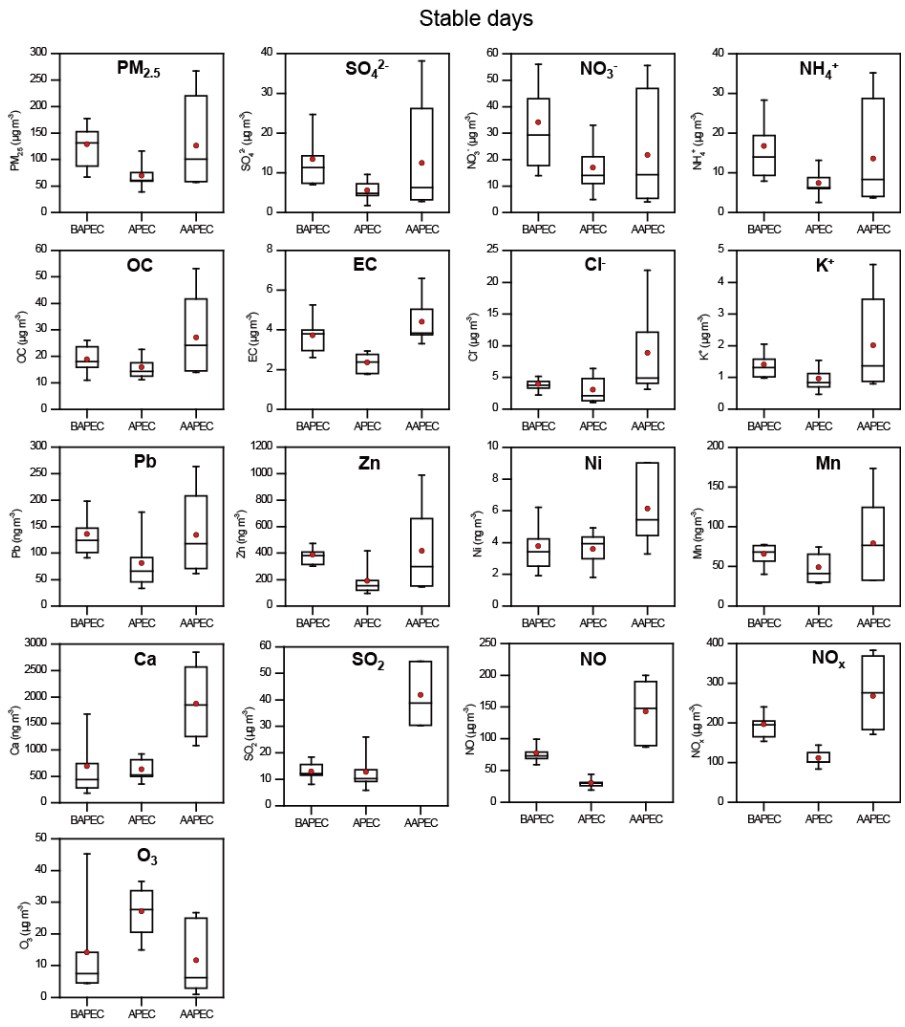

Figure 9. Variations of air pollutant concentrations during days with stable
meteorological conditions during the APEC 2014 campaign, including $PM_{2.5}$, sulphate,
nitrate, and ammonium (SNA), organic carbon (OC), elemental carbon (EC), $Cl^-$, $K^+$,
elements (Pb, Zn, Ni, Mn, and Ca), and gaseous pollutants ($SO_2$, NO, $NO_x$, and $O_3$).
The red points represent mean values. The black cross bars are median values. The
black box denotes the $25^{th}$ and $75^{th}$ percentiles. The whiskers represent the maximum
and minimum, respectively. B: before; A: after.






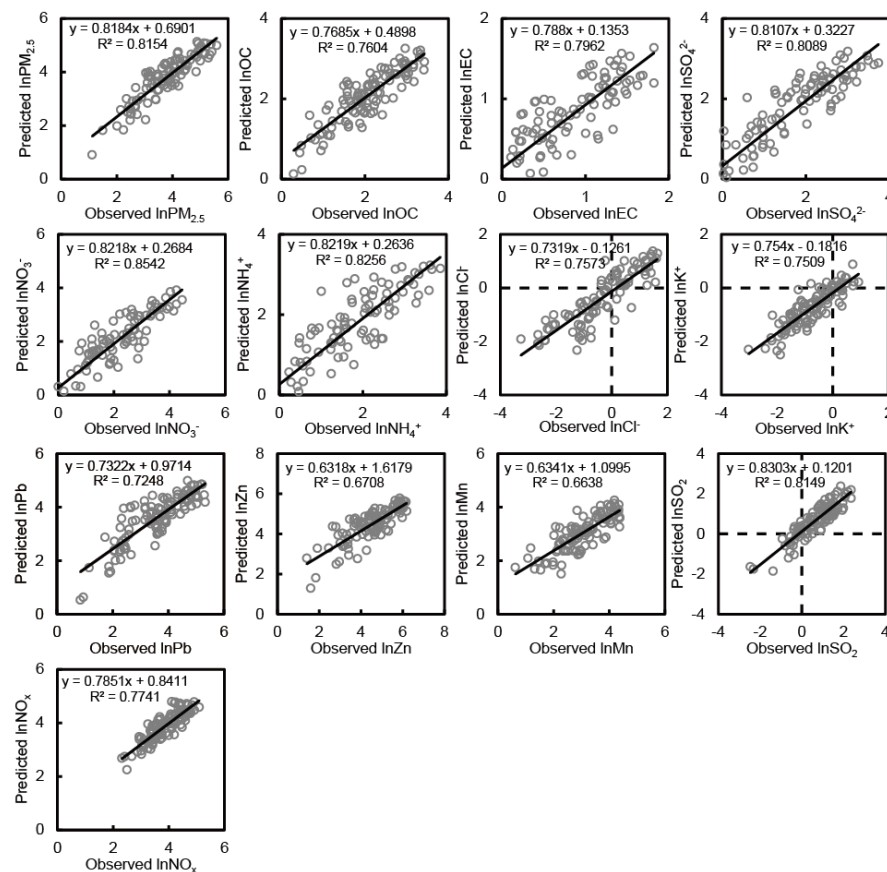


Figure 10. Scatter plot and correlations between GLM-predicted (*y*-axis) and observed

(*x*-axis) concentrations of pollutants transformed to a natural log. The linear regression

equations and $R^2$ values are given.






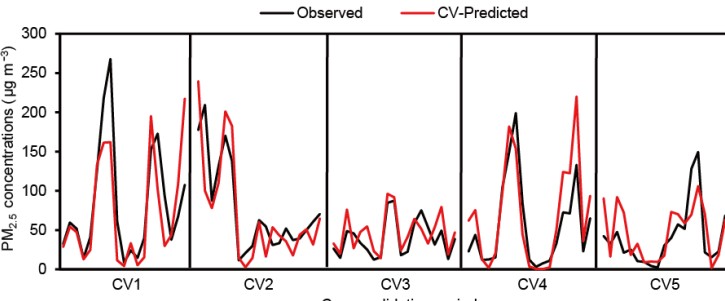


Figure 11. Time series of the observed and cross-validation (CV) predicted PM$_{2.5}$
concentrations during five CV periods. The black line represents the observed PM$_{2.5}$
concentration and the red line represents the CV-predicted PM$_{2.5}$ concentration.





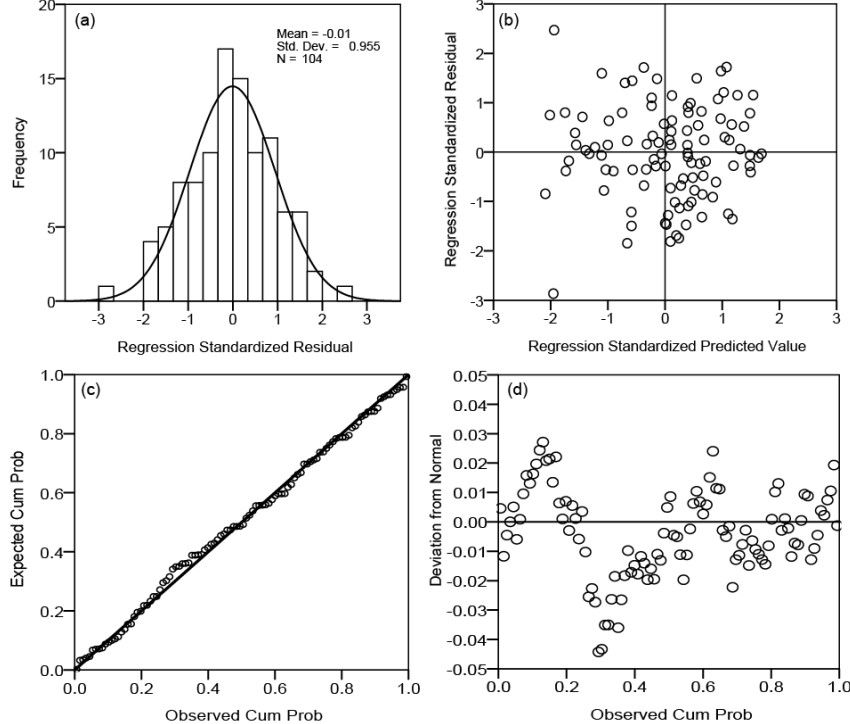


Figure 12. Residual analysis of the model. (a) Histogram of the regression standardized
residual. (b) Scatter plot between the regression standardized predicted value and
regression standardized residual. (c) Normal P-P plot of the regression standardized
residual between the observed cumulative probability and expected cumulative
probability. (d) De-trended normal P-P plot of the standardized residual of observed
cumulative probability.