# Peer review of "The Role of Meteorological Conditions and Pollution Control"

_Atmospheric Chemistry and Physics, 2017_

## Referee Comment (RC1) · Anonymous Referee #1 · 27 Jul 2017

General Comments: This manuscript well selected two periods, i.e., the APEC 2014 and Parade 2015, to investigate the meteorological conditions and pollution control strategies in reducing air pollution in Beijing. It is scientifically sound, original, well written and concise. I recommend to accept it after minor revision as indicated below.

1 Too many tables included in the main text, so I suggest the authors adjusting the structures of the manuscript and move some of the tables and figures to the supplement to make it more concise and clear. 2 The authors used two methods to separate out the influence of meteorological conditions on the air pollutant concentrations to

give a fairly and accurate evaluation of effectiveness of pollution control strategies. It seems that the authors think the GLM method is better than the "stable meteorological condition" method? If so, why the authors focused on the explanations of the results of "stable meteorological condition" method?

Detailed comments

1 The abstract is too long, please give a concise and clearly written. Line 28, delete "dramatically". Line 23, the authors state that "During the APEC 24 (1 October to 31 December 2014) and Parade (1 August to 31 December 2015) sampling periods", but in Figure 1, 4, 6 and Line 235-240, the study periods were from 18/10/2014-22/11/2014 and 01/08/2015-23/09/2015. Please give more clear and consistent definition of your research periods in your manuscript, such as during, before and after "APEC" or "Parade", "AAPEC", "APEC", "BAPEC", "AParade", "Parade" and "BParade".

2 For the "Introduction" section, I suggest the authors move Table 1 and Table 2 and some related context to the "Methods" Section. Line 55, "(2013)" is the reference citation format correct? please check the format of the references throughout the whole manuscript more carefully. Line 59 "2012 levels" to "levels of 2012". Line 71-72, Please cite some scientific literatures here instead of "(SEPB, 2010)", "(GEPB, 2009)"ïïjŇand "(CEPB, 2013)". Line 73-75, Only need to define abbreviations at their first occurrence. e.g. "APEC","Parade", "GLM" etc. Line 78, delete "control (Table1)". Line 83 "from" to "to". Line 86-90, Please rewrite these two sentences. You mean 54 % in Beijing, 26 % in Shijiazhuang, and 39 % in Tangshan. What is "the average concentration of total elements in PM2.5"? Line 92, what is "before" represent? Line 95, delete "e.g."

3 For the "Measurement and Methods", I just recommend the authors include the measurement, the research periods definition and control strategies, and the methods for the meteorological conditions separation in this section. Section 2.1, 2.2 and 2.3 can be combined, and some content in introduction and Section 3 can be moved to this part. Line 141, change to "the 4th Ring Road of Beijing". Line 2.2, why the authors

used the meteorological data from NCDC of the airport not the corresponding data from PKU site? Line 172, what is "AX105DR" represent for? Line 203-205, why define the variable WD? And what is the difference between (1) and (2). Line 208, change "Figure S2" to "Fig. S1", the tables and the figures should be labeled separately. Line 216, use the equation editor to give the proper format of the formulas.

4 For Section 3, the authors are suggested to rearranged the structure of the manuscript and give in-depth discussions of the results, not just mentioned the results. Line 235-240, move the annotations to the figure captions and keep the annotations in the Figure and the main text consistent, such as "Before APEC" means "BAPEC" in the main text? Line 245, delete "during the whole control period". Line 255-268, and Figure 2, the authors give detail explanations of the changes of the PM2.5 components for the "AAPEC", "APEC" and "BAPEC" etc., in my opinion, Figure 2 revealed a part of information of Figure 1, why the authors give this part of analysis? And why the components changes for different periods? Line 284-289, change "(SNA)/PM2.5" to "(SNA/PM2.5)". And why the proportion of SNA change like this? Line 290-296 Please give more clear and consistent definition of your research periods in your manuscript, such as during, before and after "APEC" or "Parade", "AAPEC", "APEC", "BAPEC", "AParade", "Parade" and "BParade". Line 311, add "(Fig. 1)" after sulphate information. Line 325-326 "during BParade and 326 AParade (25.7% and 20.3%, respectively)." to "during BParade (25.7%) and AParade (20.3%). Line 331-333, did the authors mean "the PBL heights during APEC and Parade were constantly high", but during these two periods, the PBL heights sometimes were low, please rewrite this sentence to give more clear statement. Section 3.2.1, Move this part to the methods section. What is the theoretical basis of this identification method? This method from previous study or developed by the authors? Did the authors combined the data of APEC and Parade, why not give the identification separately? Line 383, add "(S3)" after "Supplementary Information" Line 398-497, this part just description of the figures and lacks in-depth discussions of the results. What is Similarities and differences of the changes for different species and what caused the results?

5 Section 3.3 have structural problem, and I just recommend the authors adjust the manuscript in this section. Firstly, the authors should give a clear description of the model constructing and parameterization process (Table 8); Secondly, the authors should give the modeling results (Figure S4 should be moved to the main text) and give the validation check of the models (Figure 10-12); and then the authors can use the models to give the evaluations (this part in Section 3.3 is weak compared to the "stable meteorological condition method" and this part should be more emphasized in the manuscript). Line 548-550, "decreased by 58% and 63%" compared with what? Line 549-550, please correct the expressions like the following in the whole manuscript "the meteorological conditions and pollution control strategies contributed 30% and 28% to the reduction of the PM2.5 concentrations during APEC 2014, respectively, and 38% and 25% during Parade 2015, respectively". Did the authors mean the meteorological conditions decreased the PM2.5 concentrations by 30% and pollution control strategies decreased the PM2.5 concetration by 28%? Please check the manuscript and make more accurate statement.

Line 568 and table 10, why the sulfate increased by 44% ? The results is opposite to the "stable meteorological condition method" (Figure 9)?

6 For Figures and tables, the authors should give more accurate captions. Table 2, give the annotation of "AOD"("AOT"), "(MODIS/MISR)"(what does it mean?), Table 3 add "in this study" after "in the GLM", and clarified the minimum and maximum data is for daily or others? Table 4 give more accurate annotation of "BAPEC","APEC","AAPEC","Bparade","parade",and "AParade". Delete the ambiguous annotation "B:before;A:after".The same for other tables and Figures. Figure 1 "grey-shaded" to "blue-shaded", "Before APEC" to "BAPEC" and so on. Figure 8 What does this figure stand for? Not just give the explanations of "the black/red bars" or "the whiskers" stand for? Figure 9 delete "(SNA)" or "SNA=sulpate+nitrate+ammonium". Figure S4 move this figure to the main text and give the exactly labels of the x-axis, use the date format not just "the sampling period".

---

## Referee Comment (RC2) · Anonymous Referee #2 · 3 Aug 2017

Distinguishing the influence of the meteorological conditions and pollution control strategies on the pollutants concentrations is important for evaluations of the air pollution policies. The authors used the stable meteorological condition identification method and the GLM method to address this issue. Two cases, i.e. APEC 2014 and Parade 2015 were selected for study. Overall, the manuscript is well-written. The manuscript would be acceptable for publication in ACP if the following comments can be satisfactorily addressed.

Major Comments

1 Why the authors chose the stable meteorological condition identification method to give the evaluation first? It seems GLM method is more effective for the meteorological influence separation. Which method is focused on? If the stable meteorological condition identification method has limitations in quantifying the meteorological influences, why the authors give so many discussions on the quantifying results in this part, i.e. Line 393-436? Compared to the stable meteorological condition identification method, the GLM method mainly focused on the evaluation of the model performance and lack in-depth discussions. Furthermore, the validations of the GLM method is still weak in the manuscript, the authors just compared the model results of PM2.5 in literatures, line 551-552. Please give more in-depth analysis for the results of GLM method 2The authors are recommended to adjust the structure of the manuscript to give more clear and concise abstract and introduction. Some part of the "introduction" and "Results and discussions" can be moved to the method section. The "Results and discussions" should give more in-depth analysis without just give statement of the tables and Figures. See the following comments in detail. 3 Some annotations of the Figures and Tables should be more precise and accurate.

Detailed comments

1 Line 47-48, this sentence is confusion and misunderstanding. If "meteorological conditions and pollution control strategies contributed 30% and 28% to the reduction of the PM2.5 concentrations", is there any other reason to cause the reduction? Please rewrite sentences like this in the manuscript. 2 Line 62-63, what does you mean here? 3 Line 64-80, the authors list the special events for air pollution control, are there related studies on these events? Please add some scientific references here. 4Line 90-91, the statement here is quite obscure. Please give a clear and accurate summary of the previous studies. 5Line 95, add more references here to back your statement. 6line 130-134, the authors give the advantages of the GLM methods. "(3) in addition to predicting PM2.5 mass concentrations, our model could also predict concentrations of gaseous pollutants and individual PM2.5 components." Other methods can not predict

concentrations of gaseous pollutants and individual PM2.5 components? However, I think for most reader, they more concern about the correctness and effectiveness of the method. 7Line 162-168 Why the authors used the data from BCIA? Did the meteorological data can match with the observation data of the pollutants? 8line 183 "OCEC" to "OC/EC" 9Line201-205 Why the authors define "variable WD" and separate to (1) and (2) 10 What the physical meaning of $\beta 0$ i.e. the intercept? 11 Line233, what is the study period? 2014.10.01-2014.12.31 and 2015.08.01-2015.12.31 not match with the data shown in Figure1 12Line 255-268 what does the results imply? 13Line276-278, "indicating that OC and EC were mainly derived from the same sources during both pollution control periods, and were from different sources during the non-control periods." Why and how the sources changes? 14line280-281, why the secondary OC (SOC) formation contribution from residential solid fuel (coal and biomass) are higher in the control period? 15 Line 341-353 what is the basis for this method 16 Line 568-597 Please give in-depth discussion of the results. Why the authors use positive value to represent decrease? Why the sulfate increase during APEC?

Other Comments 1 Give the full name of abbreviations only for the first time they appear. 2 "during the APEC/Parade" can be labeled as "DAPEC/DParade" to avoid confusion of the study periods 3Table 1 give the air pollution control measures for APEC and Parade. Are the pollution control strategies different? If there is different, can use this to validate the GLM method?

---

## Author Comment (AC1) · 14 Sep 2017

**Response to the *Interactive comments from Referees on* "The Role of Meteorological Conditions and Pollution Control Strategies in Reducing Air Pollution in Beijing during APEC 2014 and Parade 2015"** *by* **Pengfei Liang et al.**

We thank the referees for the helpful comments. We have revised the manuscript according to the suggestions and responded to their concerns below.

**Referee #1:**

**General Comments:**

***General Comment 1:*** *Too many tables were included in the main text, so I suggest the authors adjust the structures of the manuscript and move some of the tables and figures to the supplement to make it more concise and clear.*

**Response to General Comment 1:** Accepted. We move a number of tables and figures to the supplement or other sections of the manuscript.

Table 2 (Table 1 at present) remains in "Introduction", since it illustrates the background of the GLM in this study. By comparing the GLM with the statistic models used in other studies, the theoretical basis and advantages of the GLM can be better illustrated.

**Changes in the Manuscript:** Table 1 (Table 2 at present) is moved to "Measurements and Methods". Table 5 (Table S5 at present) and Table 6 (Table S7 at present), Figure 5 (Figure S3 at present), Figure 6 (Figure S4 at present), Figure 9 (Figure S8 at present), and Figure 12 (Figure S9 at present)

are moved from the main text to supplement. Please refer to Page 9 Line 178-180 and Page 12 Line 237-242.

*General Comment 2: The authors used two methods to separate out the influence of meteorological conditions on the air pollutant concentrations to give a fairly and accurate evaluation of effectiveness of pollution control strategies. It seems that the authors think the GLM method is better than the "stable meteorological condition" method? If so, why the authors focused on the explanations of the results of "stable meteorological condition" method?*

**Response to General Comment 2:** Agree. The discussion of the results of the GLM method is weak and need more in-depth discussion. .

**Changes in the Manuscript:** The results of the "stable meteorological condition" method are simplified. Please refer to Page 17 Line 351-356. The results of the GLM method is emphasized in Section 3.3. Figure S4 (Figure 8 at present) "Time series of the observed and GLM-predicted pollutant concentrations" is moved from the supplement to the manuscript. Please refer to Page 18 Line 382-384. Table 8 is added to the manuscript. Please refer to Page 21 Line 450 to Page 22 Line 462. The discussion of pollutant concentrations variations is emphasized. Please refer to Page 24 Line 496-499 and Page 24 Line 506-509. Section 3.3.4 "Uncertainties of the GLM" is added to the manuscript. Please refer to Page 26 Line 544-557.

[Figure]

Figure 8. Time series of the observed (in black line) and GLM-predicted pollutant concentrations (in red line).

Table 8. The influence of the meteorological parameters included in the GLMs on pollutant concentrations[1].

| Parameters | Included in the GLM (times)[2] | $PM_{2.5}$ | EC | OC | $SO_4^{2-}$ | $NO_3^-$ | $NH_4^+$ | $Cl^-$ | $K^+$ | Pb | Zn | Mn | $SO_2$ | $NO_x$ |
|---|---|---|---|---|---|---|---|---|---|---|---|---|---|---|
| PBL | 13 | - | - | - | - | +- | - | - | +- | - | - | - | - | - |
| $WS_{(lag)}$ | 9 | - | - |  | - | - |  |  | - | - | - |  | - |  |
| $PREC_{(lag)}$ | 8 | - | - | - | - |  |  |  |  | - |  | - | - | - |
| PREC | 7 | - |  |  | - | - | - |  | - |  |  |  | - | - |
| WS | 7 | - |  | - | + | + |  |  | - | - | - |  |  |  |
| RH | 6 |  |  |  | + | + | + |  | + |  |  |  | - | - |
| $PBL_{(lag)}$ | 5 | + | + |  | + |  |  |  | - |  |  | + |  |  |
| $RH_{(lag)}$ | 5 |  |  | - |  | - | - | - | - |  |  |  |  |  |
| T | 5 |  | + | + | + | +- |  |  |  |  |  |  |  | -+ |
| $T_{(lag)}$ | 5 | + |  |  |  |  | + | - | + | + |  |  |  |  |
| $WD/WS_{(lag)}$ | 4 | + |  | + |  |  |  |  | + |  |  | + |  |  |
| SLP | 3 | - |  |  |  |  |  |  |  |  |  | - | + |  |
| WD | 3 |  |  | + |  |  | + | + |  |  |  |  |  |  |
| WD/WS | 3 |  | + |  |  |  |  |  |  |  |  | + |  | + |
| $WD_{(lag)}$ | 1 |  |  |  |  |  |  |  |  |  |  | + |  |  |

[1] "+" represents the positive correlation, and "-" represents the negative correlation between meteorological parameters and pollutant concentrations.

[2] If a parameter is included in the model for several times, it will be counted as one time.

**Detailed Comments:**

*Detailed Comment 1: The abstract is too long, please give a concise and clearly written. Line 28, delete "dramatically". Line 23, the authors state that "During the APEC (1 October to 31 December 2014) and Parade (1 August to 31 December 2015) sampling periods", but in Figure1, 4, 6 and Line235-240, the study periods were from 18/10/2014-22/11/2014 and 01/08/2015-23/09/2015. Please give more clear and consistent definition of your research periods in your manuscript, such as during, before and after "APEC" or "Parade", "AAPEC", "APEC", "BAPEC", "AParade", "Parade" and "BParade".*

**Response to Detailed Comment 1:** Accepted. "Abstract" has been simplified. For the definition of our research periods, 1) the control periods of APEC and Parade are 03/11/2014-12/11/2014 and 20/08/2015-03/09/2015; 2) the APEC/Parade campaigns consist of before, during, and after APEC/Parade, from 18/10/2014-22/11/2014 and 01/08/2015-23/09/2015; 3) the sampling periods are 01/10/2014-31/12/2014 and 01/08/2015-31/12/2015, which are used to better match the statistical model (GLM). In correspondence, we give more clear and consistent definition of our research periods in the tables and figures.

**Changes in the Manuscript:** The sentence "We therefore developed a generalized linear regression model (GLM) to establish the relationship between the concentrations of air pollutants and meteorological parameters" has been deleted. The sentence "During the APEC (1 October to 31 December 2014) and Parade (1 August to 31 December 2015) sampling periods" has been deleted. The sentence "The concentrations of all pollutants except ozone decreased dramatically (by more than 20%) during both events, compared with the levels during non-control periods" has been deleted. The sentence

"(i.e. when the daily average wind speed (WS) was less than 2.50 m s$^{-1}$ and planetary boundary layer (PBL) height was lower than 290 m)" has been deleted. The sentence " We found that the average PM$_{2.5}$ concentration during APEC decreased by 45.7% compared with the period before APEC and by 44.4% compared with the period after APEC. This difference was attributed to emission reduction efforts during APEC" has been deleted.

Section 2.2 "Research Periods Definition and Control Strategies" is added to the manuscript. Please refer to Page 8 Line 168 to Page 9 Line 177.

*Detailed Comment 2: For the "Introduction" section, I suggest the authors move Table 1 and Table 2 and some related context to the "Methods" Section. Line 55, "(2013)" is the reference citation format correct? Please check the format of the references throughout the whole manuscript more carefully. Line59"2012levels"to"levelsof2012". Line71-72, please cite some scientific literatures here instead of "(SEPB, 2010)", "(GEPB, 2009)"and "(CEPB, 2013)". Line73-75, only need to define abbreviations at their first occurrence. e.g. "APEC", "Parade", "GLM" etc. Line 78, delete "control (Table1)". Line 83 "from" to "to". Line 86-90, please rewrite these two sentences. You mean 54 % in Beijing, 26 % in Shijiazhuang, and 39 % in Tangshan. What is "the average concentration of total elements in PM2.5"? Line 92, what is "before" represent? Line 95, delete "e.g."*

**Response to Detailed Comment 2:** Accepted. "the State Council of China (2013)" is equal to "(the State Council of China, 2013)". We have checked the format of the references throughout the whole manuscript. The air pollution control measures implemented for the events come from the public documents and some scientific literatures have been added.

**Changes in the Manuscript:** The control strategies in Table 1 (Table 2 at present) are moved to "Measurements and Methods". Please refer to Page 9 Line 178-180 and Page 12 Line 237-242. "2012 levels" is changed to "levels of 2012". Please refer to Page 3 Line 50. A number of scientific literatures are added to support the air pollution control measures implemented for the events. Please refer to Page 3 Line 61-63. The abbreviations of "APEC", "Parade", and "GLM" are used after being defined at their occurrence in "Abstract". Please refer to Page 3 Line 64. The decreased ratios of the concentrations of total elements reported by Wen et al. (2016) are deleted. Please to Page 4 Line 73-75.In the study of Han et al. (2015), "before" is changed to "before APEC". Please refer to Page 4 Line 77.

*Detailed Comment 3: For the "Measurement and Methods", I just recommend the authors include the measurement, the research periods definition and control strategies, and the methods for the meteorological conditions separation in this section. Section 2.1, 2.2 and 2.3 can be combined, and some content in introduction and Section 3 can be moved to this part. Line 141, change to "the 4th Ring Road of Beijing". Section 2.2, why the authors used the meteorological data from NCDC of the airport not the corresponding data from PKU site? Line 172, what is "AX105DR" represent for? Line 203-205, why define the variable WD? And what is the difference between (1) and (2). Line 208, change "Figure S2" to "Fig. S1", the tables and the figures should be labeled separately. Line 216, use the equation editor to give the proper format of the formulas.*

**Response to Detailed Comment 3:** Accepted. The structure of "Measurement and Methods" is accordingly adjusted. We use the meteorological data from NCDC of the airport rather than the corresponding data from PKU site, because the data of WS and WD is influenced by the building northern of the observation site and these two meteorological parameters might influence the pollutant concentrations significantly. The meteorological data from NCDC are integrated and continuous, which can represent the meteorological influences on the daily average pollutant concentrations at PKU site. "AX105DR" is the instrument model of the electronic balance. The definition of "variable WD" is given by the JetStream Glossary of NOAA (http://www.srh.weather.gov/srh/jetstream/append/glossary_v.html). For the WD data of NCDC, it is invalid when the wind is calm or keeps fluctuating during a time period. Thus, the data of WD indicated with "calm and variable" are grouped into an independent category.

**Changes in the Manuscript:** The structure of "Measurement and Methods" is adjusted. "Measurements of Air Pollutants", "Meteorological Data", and "Analysis of the $PM_{2.5}$ Filter Samples" are combined into Section 2.1 "Measurements". Section 2.2 "Research Periods Definition and Control Strategies" is added. The introduction of the "stable meteorological condition" method is moved to "Measurement and Methods" and combined with the GLM method into Section 2.3 "Methods for the Meteorological Conditions Separation". Please refer to Page 6 Line 116, Page 8 Line 168 to Page 9 Line 180, and Page 9 Line 181 to Page 10 Line 196. "the fourth ring road" is replaced by "the 4th Ring Road of Beijing". Please refer to Page 6 Line 122. The citation of the JetStream Glossary of NOAA is added. Please refer to Page 10 Line 208-209.

*Detailed Comment 4: For Section 3, the authors are suggested to rearrange the structure of the manuscript and give in-depth discussions of the results, not just mentioned the results. Line 235-240, move the annotations to the figure captions and keep the annotations in the Figure and the main text consistent,*

*such as "Before APEC" means "BAPEC" in the main text? Line245, delete "during the whole control period". Line255-268, and Figure 2, the authors give detail explanations of the changes of the PM2.5 components for the "AAPEC", "APEC" and "BAPEC" etc., in my opinion, Figure 2 revealed a part of information of Figure1,why the authors give this part of analysis? And why the components changes for different periods? Line284-289, change"(SNA)/PM2.5"to"(SNA/PM2.5)". And why the proportion of SNA change like this? Line 290-296 Please give more clear and consistent definition of your research periods in your manuscript, such as during, before and after "APEC" or "Parade", "AAPEC", "APEC", "BAPEC", "AParade", "Parade" and "BParade". Line 311, add "(Fig. 1)" after sulphate information. Line 325-326 "during BParade and 326 AParade (25.7% and 20.3%, respectively)." to "during BParade (25.7%) and AParade (20.3%). Line 331-333, did the authors mean "the PBL heights during APEC and Parade were constantly high", but during these two periods, the PBL heights sometimes were low, please rewrite this sentence to give more clear statement. Section 3.2.1, Move this part to the methods section. What is the theoretical basis of this identification method? This method from previous study or developed by the authors? Did the authors combined the data of APEC and Parade, why not give the identification separately? Line 383, add "(S3)" after "Supplementary Information" Line 398-497, this part just description of the figures and lacks in-depth discussions of the results. What is Similarities and differences of the changes for different species and what caused the results?*

**Response to Detailed Comment 4:** Accepted. "BAPEC", "APEC", and "AAPEC" mean before, during, and after APEC; "BParade", "Parade", and "AParade" mean before, during, and after Parade.

We discuss the proportions of the measured components in $PM_{2.5}$ before, during, and after APEC/Parade. Although all the component concentrations decrease during APEC and Parade, the proportions of different components in $PM_{2.5}$ show different changing patterns. The proportions of OC and elements in $PM_{2.5}$ tend to increase and the proportion of SNA in $PM_{2.5}$ tends to decrease. This indicates that secondary formation of SNA from primary gaseous pollutants contributes to high pollution level significantly. During APEC/Parade, emission reduction results in decreased proportions of SNA and increased proportions of other components in $PM_{2.5}$.

The PBL heights increase on 5, 11, and 12 November during APEC and are mostly higher than 400 m during Parade. The identification of stable meteorological periods is based on the empirical and mathematical relationship between air pollution levels and both WS and PBL height. We combine the data of APEC and Parade so that different pollution levels can be included in the scattering plot for better influences of WS and PBL height on $PM_{2.5}$.

**Changes in the Manuscript:** "BAPEC", "APEC", and "AAPEC" are replaced by before, during, and after APEC; "BParade", "Parade", and "AParade" are replaced by before, during, and after Parade. Please refer to Page 13 Line 258-261, 268-270, and 272-273, Page 14 Line 300, and Page 15 Line 301, 306-308, 310-311, and 313-314."during the whole control period" is deleted. Please refer to Page 12 Line 253. "(SNA)/$PM_{2.5}$" is changed to"(SNA/$PM_{2.5}$)". Please refer to Page 14 Line 296, 299, and 300. "(Figure 1)" is added after sulphate information. Please refer to Page 15 Line 322. The figures "The prevalence of WD during the APEC and Parade campaigns" and "Time series of daily average $PM_{2.5}$ concentrations and PBL heights during the APEC and Parade campaigns" are moved to the supplement S3 and S4. "during BParade and AParade (25.7% and 20.3%, respectively)" is changed to "during BParade (25.7%) and AParade (20.3%)". The statement of the PBL heights has been rewritten. Please refer to the supplement S3 and S4. The introduction of the

"stable meteorological condition" method is moved to "Measurements and Methods". Please refer to Page 9 Line 182 to Page 10 Line 196. "(S6)" is added after "Supplementary Information". Please refer to Page 17 Line 358.

The discussion of the results of the GLM method has been improved. Figure S4 (Figure 8 at present) is moved from the supplement to the manuscript in Section 3.3.1, showing the time series of the observed pollutant and GLM-predicted pollutant concentrations. Please refer to Page 18 Line 382-384. The description of the model results is emphasized. Table 8 is added to Section 3.3.2, summarizing the meteorological parameters included in the models and their influence on pollutant concentrations. Please refer to Page 21 Line 450 to Page 22 Line 462. The changes for different pollutant concentrations are further discussed. Please refer to Page 24 Line 496-499 and 506-509. Section 3.3.4 "Uncertainties of the GLM" is added to the manuscript. Please refer to Page 26 Line 544-557.

***Detailed Comment 5:*** *Section 3.3 have structural problem, and I just recommend the authors adjust the manuscript in this section. Firstly, the authors should give a clear description of the model constructing and parameterization process (Table 8); Secondly, the authors should give the modeling results (FigureS4 should be moved to the main text) and give the validation check of the models (Figure 10-12); and then the authors can use the models to give the evaluations (this part in Section 3.3 is weak compared to the "stable meteorological condition method" and this part should be more emphasized in the manuscript). Line 548-550, "decreased by 58% and 63%" compared with what? Line 549-550, please correct the expressions like the following in the whole manuscript "the meteorological conditions and pollution control strategies contributed 30% and 28% to the reduction of the $PM_{2.5}$ concentrations during APEC 2014, respectively, and 38% and 25% during*

*Parade 2015, respectively". Did the authors mean the meteorological conditionsdecreasedthePM2.5concentrationsby30%andpollutioncontrolstrategies decreased the PM2.5 concentration by 28%? Please check the manuscript and make more accurate statement. Line 568 and table 10, why the sulfate increased by 44%? The results is opposite to the "stable meteorological condition method" (Figure 9)?*

**Response to Detailed Comment 5:** Accepted. The structure of Section 3.3 is adjusted accordingly and the results of the GLM method have been more emphasized in the manuscript. We apply the GLM to predict air pollutant concentrations during APEC and Parade based on meteorological parameters. The difference between the observed and GLM-predicted concentrations is attributed to emission reduction through the implementation of air pollution control strategies.

The concentrations of sulphate are determined by primary emissions and secondary transformation from $SO_2$; thus, the changes in sulphate concentrations may not reflect the effectiveness of emission control strategies. One needs to also include the changes in $SO_2$ concentrations by adding the concentration of total S to discussion.

**Changes in the Manuscript:** The structure of Section 3.3 is adjusted accordingly, including the model performance and cross-validation test, model description, quantitative estimates of the contribution of meteorological conditions to air pollutant concentrations, and uncertainties of the GLM. Please refer to Page 18 Line 380, Page 21 Line 430, Page 22 Line 463, and Page 26 Line 544. The assumption of the GLM is added when discussing the contributions of meteorological conditions and pollution control strategies to the reduction of pollutant concentrations. Please refer to Page 23 Line 479-480. The reduction of sulphate concentrations is discussed in the manuscript. Please refer to Page 24 Line 510 to Page 25 Line 524.

***Detailed Comment 6:*** *For figures and tables, the authors should give more accurate captions. Table 2, give the annotation of "AOD" ("AOT"), "(MODIS/MISR)" (what does it mean?), Table 3 add "in this study" after "in the GLM", and clarified the minimum and maximum data is for daily or others? Table 4 give more accurate annotation of "BAPEC", "APEC", "AAPEC", "Bparade", "parade", and "AParade". Delete the ambiguous annotation "B: before; A: after". The same for other tables and Figures. Figure 1 "grey-shaded" to "blue-shaded", "Before APEC" to "BAPEC" and so on. Figure 8, what does this figure stand for? Not just give the explanations of "the black/red bars" or "the whiskers" stand for? Figure 9 delete "(SNA)" or "SNA=sulphate + nitrate + ammonium". FigureS4 move this figure to the main text and give the exactly labels of the x-axis, use the date format not just "the sampling period".*

**Response to Detailed Comment 6:** Accepted. We give more accurate captions for figures and tables accordingly. Figure 8 (Figure 6 at present) stands for the improvements and uncertainties of the "stable meteorological condition" method, indicating that by considering only days with stable meteorological conditions, the uncertainties associated with the percentage reduction figures are reduced and the reliability of the changes of air pollutants concentrations are improved. However, uncertainties still remain.

**Changes in the Manuscript:** The annotations of "MODIS/MISR", ""MOD/MYD" are given in Table 2 (Table 1 at present). Please refer to Page 38 Line 828. "in this study" is added after "in the GLM" in Table 3. Please refer to Page 40 Line 834. More accurate annotations of "BAPEC", "APEC", "AAPEC", "Bparade", "parade", and "AParade" are given in relevant tables and figures. "B: before; A: after" is replaced by "BAPEC/BParade: before APEC/Parade, AAPEC/AParade: after

APEC/Parade". Please refer to Page 41 Line 839-840 and Page 48 Line 882-883. "grey-shaded" is changed to "blue-shaded". Please refer to Page 47 Line 875. The caption of Figure 8 (Figure 6 at present) is modified, illustrating the purpose of the figure. Please refer to Page 52 Line 914-915. Figure S4 (Figure 8 at present) is moved from the supplement to the manuscript. Please refer to Page 54 Line 929. Figure S4 (Figure 8 at present) is moved from the supplement to the main text and the date format in the labels of x-axis is given. Please refer to Page 54 Line 927-929.

**Referee #2:**

**Major Comments:**

*Major Comment 1: Why the authors chose the stable meteorological condition identification method to give the evaluation first? It seems GLM method is more effective for the meteorological influence separation. Which method is focused on?*

*If the stable meteorological condition identification method has limitations in quantifying the meteorological influences, why the authors give so many discussions on the quantifying results in this part, i.e. Line 393-436? Compared to the stable meteorological condition identification method, the GLM method mainly focused on the evaluation of the model performance and lack in-depth discussions.*

*Furthermore, the validations of the GLM method is still weak in the manuscript, the authors just compared the model results of $PM_{2.5}$ in literatures, line 551-552. Please give more in-depth analysis for the results of GLM method*

**Response to Major Comment 1:** Agree. We focus on the GLM method rather than the "stable meteorological condition" method, because the GLM method has been proved to be more effective for the meteorological influence separation. We firstly introduce the "stable meteorological condition" method for the reasons as following. The "stable meteorological condition" method and the "statistical models (e.g. GLM)" method are two major methods to help separate the meteorological influences on pollutant concentrations in the former studies. The "stable meteorological condition" method is easier to achieved and more widely applied. However, there still exists limits of the application of the "stable meteorological condition" method that the stable meteorological conditions are determined subjectively e.g. by meteorological maps and weather systems. Thus, we determine the days with stable meteorological conditions based on specific meteorological parameters of wind speed and PBL height quantitatively and evaluate the improvement of the "stable meteorological condition" method.

As a result, uncertainties of the "stable meteorological condition" method still remain in quantifying the meteorological influences, although the size of these uncertainties has been reduced. This may be due to the limited sample size on days with stable meteorological conditions during the control periods. It is therefore necessary to further quantify the meteorological influences with the GLM method. Indeed, the discussion of the results of the GLM method is weak compared to the "stable meteorological condition" method.

In fact, we give systematic validation check of the $PM_{2.5}$ GLM, including the $R^2$ values of the linear regression equations showing the correlations between GLM-predicted and observed concentrations of pollutants, and the results of the cross-validation test (Figure 10 (Figure 7 at present), Figure 11 (Figure 9 at present), and Table 7 (Table 5 at present)). As the referee suggested, the results of the GLM method have been more emphasized in the manuscript.

**Changes in the Manuscript:** The results of the "stable meteorological conditions" method is simplified in Section 3.2. Figure 5 (Figure S3 at present), Figure 6 (Figure S4 at present), and Table 5 (Table S5 at present) are moved from the main text. Please refer to Page 16 Line 331-333 and 340-342. The variations of pollutant concentrations under stable meteorological conditions in Figure 9 (Figure S8 at present) are simplified and moved from the main text. Please refer to Page 17 Line 351-356. Table 6 (Table S7 at present) is moved from the main text, listing the percentage differences among the mean $PM_{2.5}$ concentrations of four periods that are randomly selected from within the non-control days of the APEC and Parade campaigns. Please refer to Page 18 Line 368-370.

The results of GLM method is emphasized in Section 3.3. The structure of Section 3.3 is adjusted accordingly, including the model performance and cross-validation test, model description, quantitative estimates of the contribution of meteorological conditions to air pollutant concentrations, and uncertainties of the GLM. Please refer to Page 18 Line 380, Page 21 Line 430, Page 22 Line 463, and Page 26 Line 544. Figure S4 (Figure 8 at present) is moved from the supplement to the manuscript in Section 3.3.1, showing the time series of the observed pollutant and GLM-predicted pollutant concentrations. Please refer to Page 18 Line 382-384.The description of the model results is emphasized. Table 8 is added to Section 3.3.2, summarizing the meteorological parameters included in the models and their influence on pollutant concentrations. Please refer to Page 21 Line 450 to Page 22 Line 462. The changes for different pollutant concentrations are further discussed. Please refer to Page 24 Line 496-499 and 506-509. Section 3.3.4 "Uncertainties of the GLM" is added to the manuscript. Please refer to Page 26 Line 544-557.

*Major Comment 2: The authors are recommended to adjust the structure of the manuscript to give more clear and concise abstract and introduction. Some part of the "introduction" and "Results and discussions" can be moved to the method section. The "Results and discussions" should give more in-depth analysis without just give statement of the tables and Figures. See the following comments in detail.*

**Response to Major Comment 2:** Accepted. The "Abstract" and "Introduction" are modified to be clearer and more concise.

**Changes in the Manuscript:** For the "Abstract", the variations of pollutant concentrations are deleted. The results of the "stable meteorological condition" method are deleted. Please refer to Page 2 Line 23-26. For the "Introduction", Table 1 (Table 2 at present) is moved to "Measurements and Methods". Please refer to Page 9 Line 178-180. For the "Measurements and Methods", "Measurements of Air Pollutants", "Meteorological Data", and "Analysis of the $PM_{2.5}$ Filter Samples" are combined into Section 2.1 "Measurements". Section 2.2 "Research Periods Definition and Control Strategies" is added. The introductions of the "stable meteorological condition" method has been moved to "Measurement and Methods" and combined with the GLM method into Section 2.3 "Methods for the Meteorological Conditions Separation". Please refer to Page 6 Line 116, Page 8 Line 168 to Page 9 Line 180, and Page 9 Line 181 to Page 10 Line 196.

*Major Comment 3: Some annotations of the Figures and Tables should be more precise and accurate.*

**Response to Major Comment 3:** Accepted. Please refer to the following responses in detail.

**Changes in the Manuscript:** For the tables, "in the study" is added after "in the GLM" in Table 3. Please refer to Page 40 Line 834. "B:before; A: after" is replaced by "BAPEC/BParade: before APEC/Parade, AAPEC/AParade: after APEC/Parade" in Table 4. Please refer to Page 41 Line 839-840.

For the figures, "grey-shaded" is replaced by "blue-shaded" in Figure 1. Please refer to Page 47 Line 875. "B:before; A: after" is replaced by "BAPEC/BParade: before APEC/Parade, AAPEC/AParade: after APEC/Parade" in Figure 2. Please refer to Page 48 Line 882-883. "(SNA)/$PM_{2.5}$" is changed to"(SNA/$PM_{2.5}$)" in Figure 4. Please refer to Page 50 Line 894-895. "The percentage reductions of pollutant concentrations under similar meteorological conditions." is added in front of the annotation of Figure 8 (Figure 6 at present). Please refer to Page 52 Line 914-915.

**Detailed Comments:**

*Detailed Comment 1:* Line 47-48, this sentence is confusion and misunderstanding. If "meteorological conditions and pollution control strategies contributed 30% and 28% to the reduction of the $PM_{2.5}$ concentrations", is there any other reason to cause the reduction? Please rewrite sentences like this in the manuscript.

**Response to Detailed Comment 1:** Agree. The concentrations of air pollutants could be influenced by meteorological parameters, emission intensities, and chemical transformation. In our study, the results of the GLM method are based on the assumption that the pollutant concentrations are only the function of meteorological conditions and emission intensities during the control periods of APEC and Parade.

**Changes in the Manuscript:** The assumption that "the concentrations of air pollutants are only determined by meteorological conditions and emission intensities" is added after the reduction of the $PM_{2.5}$ concentrations. Please refer to Page 2 Line 38-39 and Page 23 Line 479-480.

*Detailed Comment 2:* Line 62-63, what do you mean here?

**Response to Detailed Comment 2:** The long-term strategies cannot improve the air quality in the short term, thus it is difficult to evaluate the effectiveness of these strategies.

**Changes in the Manuscript:** "in the short term" is added after the sentence. Please refer to Page 3 Line 55.

*Detailed Comment 3: Line64-80, the authors list the special events for air pollution control, are there related studies on these events? Please add some scientific references here.*

**Response to Detailed Comment 3:** Accepted. There are related studies on these events and the references have been added.

**Changes in the Manuscript:** "Huang et al., 2012" and "Liu et al., 2013" are added after the 41$^{st}$ Shanghai World Expo in 2010 and the 16$^{th}$ Guangzhou Asian Games and Asian Para Games in 2010. Please refer to Page 3 Line 61-63, Page 30 Line 659-662, and 665-667.

*Detailed Comment 4: Line 90-91, the statement here is quite obscure. Please give a clear and accurate summary of the previous studies.*

**Response to Detailed Comment 4:** Accepted. The decreased ratios of the extinction and absorbance coefficients during APEC are compared with those before APEC.

**Changes in the Manuscript:** "before APEC" is added after the sentence. Please refer to Page 4 Line 77.

***Detailed Comment 5:*** *Line 95, add more references here to back your statement.*

**Response to Detailed Comment 5:** Accepted. More references are added.

**Changes in the Manuscript:** "Calkins et al., 2016" is added. Please refer to Page 4 Line 80 and Page 29 Line 606-609.

***Detailed Comment 6:*** *Line 130-134, the authors give the advantages of the GLM methods. "(3) in addition to predicting PM$_{2.5}$ mass concentrations, our model could also predict concentrations of gaseous pollutants and individual PM$_{2.5}$ components." Other methods cannot predict concentrations of gaseous pollutants and individual PM$_{2.5}$ components? However, I think for most reader, they more concern about the correctness and effectiveness of the method.*

**Response to Detailed Comment 6:** The concentrations of PM$_{2.5}$, gaseous pollutants, and individual components can be predicted precisely because of the corresponding models for different pollutants, solely based on meteorological conditions.

**Changes in the Manuscript:** "by corresponding models for different pollutants" is added after the sentence. Please refer to Page 12 Line 242.

***Detailed Comment 7:*** *Line 162-168, why the authors used the data from BCIA? Did the meteorological data can match with the observation data of the pollutants?*

**Response to Detailed Comment 7:** We use the meteorological data from NCDC of the airport rather than the corresponding data from PKU site, because the data of WS and WD may be influenced by the building northern of the observation site and these two meteorological parameters can influence the pollutant concentrations significantly. The meteorological data from NCDC are integrated and continuous, which can represent the meteorological influences on the daily average pollutant concentrations at PKU site.

**Changes in the Manuscript:** Please refer to the response of the comment.

*Detailed Comment 8: Line 183, "OCEC" to "OC/EC".*

**Response to Detailed Comment 8:** Accepted.

**Changes in the Manuscript:** "OCEC" has been changed to "OC/EC". Please refer to Page 8 Line 161.

*Detailed Comment 9: Line201-205, why the authors define "variable WD" and separate to (1) and (2).*

**Response to Detailed Comment 9:** The definition of "variable WD" is given by the JetStream Glossary of NOAA (http://www.srh.weather.gov/srh/jetstream/append/glossary_v.html) rather than given by our study. For the WD data of NCDC, it is invalid when the wind is calm or keeps fluctuating during a time period. Thus, the data of WD indicated with "calm and variable" are grouped into an independent category.

**Changes in the Manuscript:** The statement of "the JetStream Glossary of NOAA" is added. Please refer to Page 10 Line 208-209.

**Detailed Comment 10:** *What the physical meaning of $\beta_0$ i.e. in the intercept?*

**Response to Detailed Comment 10:** The non-linear functions are natural log transformed and introduced into the GLM, then the coefficients in the non-linear functions are transformed into $\beta$, including $\beta_0$, $\beta_{1k}$, $\beta_{2k}$, $\beta_{3k}$, and $\beta_{4k}$, which represent the non-linear relationships between meteorological parameters and pollutant concentrations.

**Changes in the Manuscript:** Please refer to the response of the comment.

**Detailed Comment 11:** *Line 233, what is the study period? 2014.10.01-2014.12.31 and 2015.08.01-2015.12.31 not match with the data shown in Figure 1.*

**Response to Detailed Comment 11:** For the definition of our research periods, 1) the control periods of APEC and Parade are 03/11/2014-12/11/2014 and 20/08/2015-03/09/2015; 2) the APEC/Parade campaigns consist of before, during, and after APEC/Parade, from 18/10/2014-22/11/2014 and 01/08/2015-23/09/2015; 3) the sampling periods are 01/10/2014-31/12/2014 and 01/08/2015-31/12/2015, which are used to better match the statistical model (GLM). In correspondence, we give clearer and more consistent definition of our research periods in the tables and figures.

**Changes in the Manuscript:** Section 2.2 "Research Periods Definition and Control Strategies" is added. Please refer to Page 8 Line 168 to Page 9 Line 177.

**Detailed Comment 12:** *Line 255-268, what does the results imply?*

**Response to Detailed Comment 12:** The results imply the similarities of the chemical characteristics during these two events that the proportions of OC and elements in PM$_{2.5}$ tend to increase and the proportion of SNA in PM$_{2.5}$ tends to decrease during APEC and Parade.

**Changes in the Manuscript:** Please refer to the response of the comment.

*Detailed Comment 13:* Line 276-278, "indicating that OC and EC were mainly derived from the same sources during both pollution control periods, and were from different sources during the non-control periods." Why and how the sources changes?

**Response to Detailed Comment 13:** Li et al. (2017) reported that the residential burning of coal and open and domestic combustion of wood and crop residuals could contribute to more than 50% of total organic aerosol of the North China Plain during winter. During the control periods, it might be difficult to fully control the emission of residential burning. The sources of OC and EC remain to be proved by referring to more studies researching into the source apportionment and distribution during these two events.

**Changes in the Manuscript:** The citation of "Li et al., 2017" is added to support the discussion of the sources of OC. Please refer to Page 14 Line 286-290 and Page 30 Line 668-672.

*Detailed Comment 14:* Line 280-281, why the secondary OC (SOC) formation contribution from residential solid fuel (coal and biomass) are higher in the control period?

**Response to Detailed Comment 14:** The control periods of APEC and Parade are 03/11/2014-12/11/2014 and 20/08/2015-03/09/2015, mainly during early winter and early autumn respectively. During the control periods, it might be difficult to fully control the emission of residential burning. Residential solid fuel, like bulk coal and biomass burning might contribute to the higher level of SOC formation (Liu et al., 2016).

**Changes in the Manuscript:** Please refer to the response of the comment.

*Detailed Comment 15: Line 341-353, what is the basis for this method?*

**Response to Detailed Comment 15:** The identification of stable meteorological periods is based on the empirical and mathematical relationship between air pollution levels and both WS and PBL height. The meteorological parameters of WS and PBL height are decided by the scatter plot and correlation between $PM_{2.5}$ concentrations and meteorological parameters shown in Figure S2.

**Changes in the Manuscript:** Please refer to the response of the comment.

*Detailed Comment 16: Line 568-597, please give in-depth discussion of the results. Why the authors use positive value to represent decrease? Why the sulfate increase during APEC?*

**Response to Detailed Comment 16:** Agree. The discussion of the results of the GLM method is weak compared to the "stable meteorological condition" method and it has been more emphasized in the manuscript.

We apply the GLM to predict air pollutant concentrations during APEC and Parade based on meteorological parameters. The difference between the observed and GLM-predicted concentrations, which is the positive value, is attributed to emission reduction through the implementation of air pollution control strategies.

The concentrations of sulphate are determined by primary emissions and secondary transformation from $SO_2$; thus, the changes in sulphate concentrations may not reflect the effectiveness of emission control strategies. One needs to also include the changes in $SO_2$ concentrations by adding the concentration of total S to discussion.

**Changes in the Manuscript:** The results of GLM method is emphasized in Section 3.3. The structure of Section 3.3 is adjusted accordingly, including the model performance and cross-validation test, model description, quantitative estimates of the contribution of meteorological conditions to air pollutant concentrations, and uncertainties of the GLM. Please refer to Page 18 Line 380, Page 21 Line 430, Page 22 Line 463, and Page 26 Line 544. Figure S4 (Figure 8 at present) is moved from the supplement to the manuscript in Section 3.3.1, showing the time series of the observed pollutant and GLM-predicted pollutant concentrations. Please refer to Page 18 Line 382-384. The description of the model results has been emphasized. Table 8 is added to Section 3.3.2, summarizing the meteorological parameters included in the models and their influence on pollutant concentrations. Please refer to Page 21 Line 450 to Page 22 Line 462. The changes for different pollutant concentrations are further discussed. Please refer to Page 24 Line 496-499, 506-509. Section 3.3.4 "Uncertainties of the GLM" is added to the manuscript. Please refer to Page 26 Line 544-557.

**Reference**

[revised manuscript text omitted]